# Directed differentiation of functional corticospinal-like neurons from endogenous SOX6+/NG2+ cortical progenitors

Abdulkadir Ozkan[1†], Hari K Padmanabhan[1†], Seth L Shipman[1‡], Eiman Azim[1§], Priyanka Kumar[1], Cameron Sadegh[1#], A Nazli Basak[2], Jeffrey D Macklis[1]*

[1]Department of Stem Cell and Regenerative Biology, and Center for Brain Science, Harvard University, Cambridge, United States; [2]Koç University, School of Medicine Translational Medicine Research Center, Istanbul, Turkiye

*For correspondence: jeffrey_macklis@harvard.edu

[†]These authors contributed equally to this work

Present address: [‡]Gladstone Institute of Data Science and Biotechnology, San Francisco, United States; [§]Molecular Neurobiology Laboratory, Salk Institute for Biological Studies, La Jolla, United States; [#]Department of Neurological Surgery, University of California Davis, Sacramento, United States

Competing interest: The authors declare that no competing interests exist.

## eLife Assessment

This study presents **fundamental** new findings introducing a new approach for the reprogramming of brain glial cells to corticospinal neurons. The data is highly **compelling**, with multiple lines of evidence demonstrating the success of this new assay. These exciting findings set the stage for future studies of the potential of these reprogrammed cells to form functional connections *in vivo* and their utility in clinical conditions where corticospinal neurons are compromised.

**Abstract** Corticospinal neurons (CSN) centrally degenerate in amyotrophic lateral sclerosis (ALS), along with spinal motor neurons, and loss of voluntary motor function in spinal cord injury (SCI) results from damage to CSN axons. For functional regeneration of specifically affected neuronal circuitry *in vivo*, or for optimally informative disease modeling and/or therapeutic screening *in vitro*, it is important to reproduce the type or subtype of neurons involved. No such appropriate *in vitro* models exist with which to investigate CSN selective vulnerability and degeneration in ALS, or to investigate routes to regeneration of CSN circuitry for ALS or SCI, critically limiting the relevance of much research. Here, we identify that the HMG-domain transcription factor *Sox6* is expressed by a subset of NG2+ endogenous cortical progenitors in postnatal and adult cortex, and that *Sox6* suppresses a latent neurogenic program by repressing proneural *Neurog2* expression by progenitors. We FACS-purify these progenitors from postnatal mouse cortex and establish a culture system to investigate their potential for directed differentiation into CSN. We then employ a multi-component construct with complementary and differentiation-sharpening transcriptional controls (activating *Neurog2*, *Fezf2*, while antagonizing *Olig2* with *VP16:Olig2*). We generate corticospinal-like neurons from SOX6+/NG2+ cortical progenitors and find that these neurons differentiate with remarkable fidelity compared with corticospinal neurons in vivo. They possess appropriate morphological, molecular, transcriptomic, and electrophysiological characteristics, without characteristics of the alternate intracortical or other neuronal subtypes. We identify that these critical specifics of differentiation are not reproduced by commonly employed *Neurog2*-driven differentiation. Neurons induced by *Neurog2* instead exhibit aberrant multi-axon morphology and express molecular hallmarks of alternate cortical projection subtypes, often in mixed form. Together, this developmentally-based directed differentiation from cortical progenitors sets a precedent and foundation for *in vitro* mechanistic and therapeutic disease modeling, and toward regenerative neuronal repopulation and circuit repair.

## Introduction

Whether toward functional regeneration of specifically affected neuronal circuitry in disorders of the central nervous system *in vivo*, or for appropriate disease modeling and/or therapeutic screening *in vitro*, reliable approaches to accurately differentiate specific types of affected and relevant neurons are required. Overly broad classes of generic or only regionally similar neurons do not adequately reflect the selective vulnerability of neuronal subtypes in most human neurodegenerative or acquired disorders. Molecular and therapeutic findings using broad or only regionally linked classes of neurons not affected in the disorder of interest are frequently not applicable for the neurons centrally involved.

Extraordinarily diverse neurons across the nervous system, in particular within the cerebral cortex, display many distinctive features, including cellular morphology, laminar and anatomical position, patterns of input and output connectivity, cardinal molecular identifiers, electrophysiology, neurochemical properties, and ultimately their functional roles (*Fishell and Rudy, 2011*; *Greig et al., 2013*; *Harris and Shepherd, 2015*; *Ramón y Cajal, 1995*; *Sugino et al., 2006*; *Tasic et al., 2018*; *Veeraraghavan et al., 2024*). Diversity exists not only between broad cell types (e.g. excitatory projection neurons vs. inhibitory interneurons; intratelencephalic vs. cortical output ('corticofugal;' projecting away from cortex) neurons; ipsilateral associative vs. commissural), but even within seemingly homogenous populations of neurons. For example, striking and sharp molecular, connectivity, and functional distinctions exist between both spatially separated subsets and interspersed subsets of CSN, with each molecularly distinct neuronal subpopulation programmed to project to distinct segments of the spinal cord, innervate topographically distinct gray matter areas, and synapse onto distinct subsets of interneurons (*Sahni et al., 2021b*; *Sahni et al., 2021a*; *Itoh et al., 2023*). Importantly, these diverse segmentally specific subsets have selective vulnerability and/or involvement in distinct human disorders (*Sahni et al., 2020*).

Such selective involvement reflects differences between specific neuronal subtypes in their molecular regulation during development and/or maturity. Specific subtypes of neurons are thus affected in distinct developmental, neurodegenerative, and acquired disorders of the central nervous system (CNS), typically resulting in irreversible functional deficits (*Saxena and Caroni, 2011*; *Durak et al., 2022*). Particularly relevant to the work presented here, corticospinal neurons (CSN; sometimes termed 'upper motor neurons,' UMN) centrally degenerate in amyotrophic lateral sclerosis (ALS) and other motor neuron diseases, along with spinal cord 'lower motor neurons,' of entirely different developmental origin and function. Furthermore, loss of voluntary and skilled motor function in spinal cord injury results from damage to CSN axons in the corticospinal tract (*Rösler et al., 2000*; *Hains et al., 2003*).

Notably, no appropriate *in vitro* models exist with which to investigate CSN/UMN selective vulnerability and degeneration in ALS, critically limiting the relevance of much research. In contrast, the availability of useful *in vitro* models of at least immature spinal motor neurons has enabled substantial success in the spinal muscular atrophy (SMA) field, with both modeling and therapeutics (for more detailed discussion, see *Sances et al., 2016*).

Importantly, and in parallel to *in vitro* modeling, one potential regenerative approach for neurodegenerative or acquired disorders is to restore elements of the affected circuitry with new neurons that are engineered to re-establish circuit-appropriate input and output connectivity (*Qian et al., 2020*; *Czupryn et al., 2011*; *Wuttke et al., 2018*). Previous studies have demonstrated that active and quiescent progenitors exist in restricted regions of the adult brain (*Rietze et al., 2001*; *Lois and Alvarez-Buylla, 1993*; *Kuhn et al., 1997*; *Reynolds and Weiss, 1992*), and that new neurons can integrate into preexisting neural circuitry, supporting the feasibility of cellular repair in the CNS (*Czupryn et al., 2011*; *Kempermann et al., 2015*; *Feliciano et al., 2015*; *Magavi et al., 2000*; *Brill et al., 2009*; *Ohira et al., 2010*; *Chen et al., 2004*). Although transplantation of *in vitro* generated neurons, either from pluripotent stem cells (PSC) or from other developmentally distant cell types, is one potential approach (*Michelsen et al., 2015*), either *ex vivo* directed differentiation or *in situ* generation of type- or subtype-specific neurons from optimally appropriate, regionally specified resident progenitors offers several advantages. First, either approach is potentially more likely to recapitulate appropriate neuronal identity than pluripotent stem cell approaches, since presumptive partially fate-restricted resident progenitors and the desired neurons share common developmental lineage, originate from the same neural progenitor domains, and were exposed to the same diffusible and local signaling during embryonic development, thus are likely to share

significant epigenomic and transcriptomic commonality (*Roessler et al., 2014*; *Treutlein et al., 2016*; *Cahoy et al., 2008*). Avoiding transplantation via *in situ* neurogenesis would offer the additional advantage of circumventing the requirement for new neurons to migrate long distances to their sites of ultimate incorporation from an injection site with favorable local growth conditions, potentially enabling desired integration of newly recruited neurons at the single-cell level (*Wuttke et al., 2018*; *Michelsen et al., 2015*; *Espuny-Camacho et al., 2013*), emulating endogenous adult neurogenesis (*Gage, 2019*; *Bond et al., 2015*; *Kempermann, 2016*); and avoiding pathological heterotopias.

Substantial progress has been made in efforts to reprogram reactive glia *in vitro* and *in vivo* to acquire some form of neuronal identity (*Gascón et al., 2016*; *Rivetti di Val Cervo et al., 2017*; *Wu et al., 2020*; *Heinrich et al., 2014*; *Torper et al., 2015*; *Grande et al., 2013*; *Niu et al., 2013*; *Heinrich et al., 2010*; *Felske et al., 2023*; *Herrero-Navarro et al., 2021*). However, functional repair of specific circuitry requires highly directed differentiation of specific neuronal subtypes (beyond a generic neurotransmitter identity, e.g.), so new neurons can form circuit-appropriate input and output connectivity (*Mattugini et al., 2019*). Work from our lab and others have advanced this goal by identifying central molecular programs that first broadly, then increasingly precisely, control and regulate specification, diversity, and connectivity of specific cortical projection neuron subtypes during the period of their differentiation (*Greig et al., 2013*; *Veeraraghavan et al., 2024*; *Sahni et al., 2021b*; *Sahni et al., 2021a*; *Arlotta et al., 2005*; *Lodato and Arlotta, 2015*; *Ozkan et al., 2020*; *Shibata et al., 2015*; *Nord et al., 2015*; *Taverna et al., 2014*; *Srinivasan et al., 2012*; *Lui et al., 2011*; *O'Leary et al., 2007*; *Greig et al., 2016*; *Woodworth et al., 2016*; *Galazo et al., 2016*; *Galazo et al., 2023*). According to an emerging model, complementary and exclusionary sets of proneural and class-, type-, and subtype-specific transcriptional controls act in a subtype-, stage-, and dose-dependent manner to direct distinct projection neuron differentiation trajectories, while repressing alternative fates (*Ozkan et al., 2020*). This sharpens subtype identities and distinctions.

In the work reported here, we build on prior work from our lab (*Azim et al., 2009a*) identifying *Sox6* as a unique stage-specific, combined temporal and spatial, control over all cortical projection neuron development that is both expressed by all cortical-pallial/excitatory projection neuron progenitors and excluded from subpallial/interneuron progenitors, and that effectively represses the transcriptional expression of the proneural gene neurogenin 2 (*Neurog2*). We identify that a subset of NG2+ (Nerve-Glial antigen 2 is a transmembrane chondroitin sulfate proteoglycan, with the protein component encoded by the gene *Cspg4*) endogenous cortical progenitors continue to express *Sox6*, which continues to repress *Neurog2* expression and neuronal differentiation. We take advantage of genetic access to FACS-purify these endogenous cortical progenitors and establish a culture system to investigate the potential for their directed differentiation into cortical output neurons, the type of clinically relevant neurons that centrally includes CSN.

We then synthesized and applied a multi-component gene expression construct with complementary and differentiation-sharpening transcriptional controls (activating *Neurog2* and *Fezf2*, while antagonizing *Olig2* with *VP16:Olig2*) to these purified, partially fate-restricted progenitors from postnatal mouse cortex. We find that this approach directs highly specific acquisition of many cardinal morphological, molecular, and functional characteristics of endogenous corticospinal neurons, and not of the alternative intracortical or other CNS neuronal subtypes. We further investigate these results in several directions, finding, e.g., that *Neurog2* alone is not sufficient to induce a specific neuronal identity; that neurons induced by *Neurog2* instead exhibit aberrant multi-axon morphology and express molecular hallmarks of alternate cortical projection subtypes, often in mixed form.

As a proof of concept, we employ synthetically modified RNAs to control timing and dosage of the exogenous transcription factors, finding that a single pulse of *Neurog2* combined with *Fezf2* induces projection neuron differentiation from cultured SOX6+/NG2+ endogenous cortical progenitors, further highlighting the seemingly 'poised' and already partially cortical neuron fate-directed potential of these specialized progenitors. Our results demonstrate the feasibility of achieving molecularly directed, subtype-specific neuronal differentiation from a widely distributed endogenous progenitor population, with significant implications for both *in vitro* disease modeling and efforts toward therapeutic *in situ* repopulation of degenerated or injured cortical circuitry.

## Results

### Identification of SOX6+/NG2+ cortical progenitors in postnatal and adult neocortex

Progenitors and glia in postnatal and adult cortex share a common ancestry with cortical neurons (*Elsherbiny and Dobreva, 2021*). Therefore, we hypothesized that at least some of these progenitors and glia might have dormant neurogenic potential, and that a subset might have molecular characteristics that might enable their enhanced and potentially appropriate differentiation into cortical projection neurons (*Elsherbiny and Dobreva, 2021*; *Zhang et al., 2014*).

To identify this potential subset, we labeled proliferative cells in postnatal and adult cortex with an injection of BrdU (see Methods), and immunolabeled for PAX6, TBR2, SOX6, and FEZF2 –transcriptional controls that play key roles in embryonic pallial progenitors (*Greig et al., 2013*; *Hevner, 2006*; *Woodworth et al., 2012*). This experiment revealed that many BrdU+ proliferative cells continue to express SOX6 in postnatal and adult mouse cortex (*Figure 1A*, *Figure 1—figure supplement 1*). *Sox6* controls molecular segregation of dorsal and ventral telencephalic progenitors during telencephalon parcellation in important part by blocking ectopic proneural gene expression by pallial progenitors and subpallial mantle zones (*Azim et al., 2009a*). To investigate whether *Sox6* has parallel function in postnatal proliferative cells, we investigated proneural gene expression in *Sox6* null brains. Strikingly, the proneural gene *Neurog2* is ectopically expressed throughout *Sox6*-null cortex at postnatal day 6 (P6) (*Figure 1B*, *Figure 1—figure supplement 2*). This result indicates that a subset of postnatal cortical progenitors maintains latent neurogenic programs that are actively suppressed by *Sox6*, similar to its function in embryonic progenitors.

We then focused the investigation on SOX6+ cells by immunocytochemistry (ICC) and by using genetically labeled progenitors (NG2-DsRed) (*Zhu et al., 2008a*). We identify that SOX6+ cells are a subset of NG2-proteoglycan-expressing proliferative cells resident across the CNS (*Figure 1C*). These data indicate that at least a subset of SOX6+/NG2+ progenitors resident in the neocortex possess some level of dormant neurogenic competence, which might be activated with relatively focused molecular manipulation. Therefore, we targeted SOX6+/NG2+ progenitors for directed differentiation into clinically relevant cortical output neurons, including CSN.

### Purification and culture of SOX6+/NG2+ cortical progenitors

We established a culture system of purified SOX6+/NG2+ cortical progenitors to evaluate candidate transcriptional regulators for their ability to direct differentiation of SOX6+/NG2+ progenitors into cortical output neurons *in vitro*, thus enabling rigorous and iterative experimentation under controlled conditions. We used a transgenic NG2-DsRed mouse line (*Figure 1D*, *Figure 1—figure supplement 3A–C*; *Zhu et al., 2008a* to isolate DsRed-positive cells by FACS from micro-dissected dorso-lateral neocortex at P2-P6 (*Figure 1*, *Figure 1—figure supplement 3D*). Three distinct DsRed populations were identified based on fluorescence intensity: 'DsRed-Bright' (2–5%), 'DsRed-Dim' (~20%), 'DsRed-negative' (~75%) (*Figure 1E*). Quantitative PCR (qPCR) (n=4) and ICC (n=2) revealed that DsRed-Bright cells are progenitors with high expression of *Sox6*, *Cspg4 (NG2)*, and *Olig2* (*Figure 1F*), whereas DsRed-Dim cells are a heterogeneous population that includes GFAP+ astrocytes, NESTIN+ progenitors, and a subset of NG2+ progenitors (*Figure 1—figure supplement 3E–G*). To further investigate these DsRed populations, we performed RNA-seq on acutely sorted DsRed-Bright, DsRed-Dim, and DsRed-negative populations (n=5–6), and evaluated expression of a focused set of 500 genes most enriched in major cortical cell lineages (*Supplementary file 1*; *Zhang et al., 2014*). Cortical NG2+ progenitor-enriched genes are highly expressed by the DsRed-Bright population (*Figure 1G*, *Figure 1—figure supplement 4A*), whereas neuronal, astroglial, and microglial genes are depleted (*Figure 1—figure supplement 4B–D*). Together, these data indicate that DsRed-Bright cells are canonical SOX6+/NG2+ progenitors, potentially optimally suited for use in subsequent directed differentiation experiments.

We FACS-purified DsRed-Bright SOX6+/NG2+ progenitors with stringent gating and cultured them for 5 days (days-in-vitro, DIV) until they reached optimal confluency for transfection (*Figure 1H*). To promote the preservation of endogenous progenitor characteristics in culture, we performed a pilot experiment varying morphogen composition to broadly optimize serum-free medium formulation based on previously published protocols (*Figure 1—figure supplement 3H*; *Lyssiotis et al., 2007*). When cultured in this medium, progenitors proliferate robustly in response to the mitogens

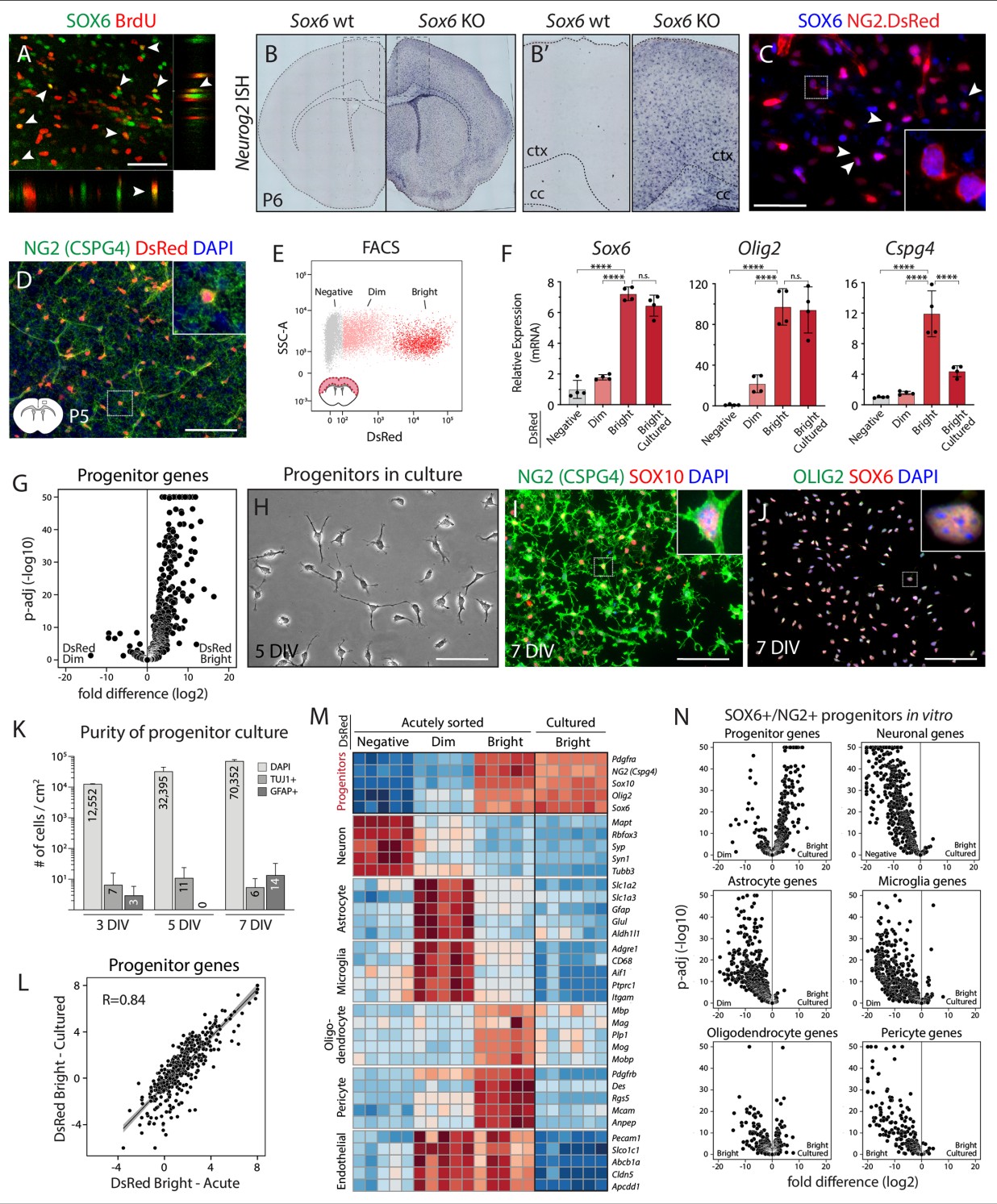

**Figure 1.** Identification and culture of SOX6+/NG2+ cortical progenitors with high purity and fidelity. (**A**) Confocal micrograph of mouse neocortex at postnatal day 7 (P7) showing expression of SOX6 by a subset of BrdU+ proliferative cells. See also *Figure 1—figure supplement 1*. (**B**) *In situ* hybridization of *Neurog2* in *Sox6* wild-type (wt) (left) and knockout (KO) (right) cortex at P6. (**B'**) Insets showing the boxed areas in B. Loss of *Sox6* results in widespread ectopic expression of *Neurog2* (**B'**). See also *Figure 1—figure supplement 2*. (**C**) Immunofluorescence showing expression of SOX6 by DsRed+ NG2+ progenitors in cortex at P5. (**D**) Immunostaining of NG2 proteoglycan in NG2-DsRed cortex at P5 shows expression of DsRed by NG2+ progenitors. Inset: a cell with a strong DsRed signal in the cell body and NG2 proteoglycan around the main cell body and in cellular processes. See also *Figure 1—figure supplement 3A–C*. (**E**) Representative FACS plot of neocortical cells from NG2-DsRed transgenic cortex showing distinct DsRed-Bright, -Dim, and -negative populations. (**F**) qPCR analysis of *Sox6*, *Olig2*, and *Cspg4* from acutely sorted DsRed-Negative, -Dim, and -Bright

*Figure 1 continued*

populations, as well as cultured DsRed-Bright cells (5DIV), demonstrates that SOX6+/NG2+ progenitors are enriched in DsRed-Bright population and maintain key gene expression *in vitro*. See also *Figure 1—figure supplement 3D–G*. Data are presented as mean ± SD, n=4, biological replicates, *Actb* normalized data relative to DsRed-negative population. \*\*\*\**p*<0.0001, *p*≥0.05, no statistically significant difference (n.s.); ANOVA Tukey's post hoc test. (**G**) Volcano plot comparing fold difference in average expression of progenitor genes between acutely sorted DsRed-Bright and -Dim populations (RNA-seq, n=5, biological replicates). See also *Figure 1—figure supplement 4A*. (**H**) Representative brightfield image of cultured SOX6+/NG2+ (DsRed-Bright) progenitors at 5 DIV showing preserved progenitor multipolar morphology. See also *Figure 1—figure supplement 3J–P*. (**I, J**) Cultured progenitors continue expressing the key progenitor-specific molecules NG2, SOX10 (**I**), OLIG2, and SOX6 (**J**) at 7 DIV. (**K**) Quantification of TUJ1+ and GFAP+ cells at 3-, 5-, and 7 DIV shows essentially no contaminant cells in culture. Data are presented as mean ± SD, n=2, biological replicates. See also *Figure 1—figure supplement 3Q*. (**L**) Pearson correlation analysis of progenitor genes shows high similarity between acutely sorted and cultured SOX6+/NG2+ (DsRed-Bright) progenitors (*R*=0.84, *p*<2.2e-16). Data points represent log2 fold differences in gene expression relative to acutely sorted DsRed-Dim population. See also *Figure 1—figure supplement 4A*. (**M**) Heatmap of the top five marker genes for seven major cell types in brain shows that SOX6+/NG2+ progenitors are enriched in DsRed-Bright populations and that progenitor cultures are free of potential contaminants. Counts are variance-stabilizing transformed (vst) normalized data in log2 scale. (**N**) Volcano plot comparing fold differences in average expression of the top 500 genes for major cell types between cultured SOX6+/NG2+ (DsRed-Bright) progenitors and acutely sorted cells. n=5/6, biological replicates. See also *Figure 1—figure supplement 4*. Scale bars (**A, C, H**) 50 µm; (**D, I, J**) 100 µm. cc: corpus callosum, ctx: cortex.

The online version of this article includes the following figure supplement(s) for figure 1:

**Figure supplement 1.** Identification of SOX6+ cortical progenitors in postnatal and adult neocortex.

**Figure supplement 2.** Loss of Sox6 function ectopically upregulates Neurog2 throughout the forebrain.

**Figure supplement 3.** Characterization, FACS isolation, and culture of cortical SOX6+/NG2+ progenitors.

**Figure supplement 4.** SOX6+/NG2+ progenitors maintain molecular characteristics *in vitro*, and cultures are free of DsRed+ pericytes.

PDGF-A and FGF2 (*Figure 1—figure supplement 3I–N*). They maintain their cardinal molecular hallmarks, including expression of SOX6, NG2, OLIG2, and SOX10 (*Figure 1F, I, J*, *Figure 1—figure supplement 3J-M*), and conserve characteristic branched morphology with non-overlapping territorial processes (*Figure 1—figure supplement 3N–P*; *Hughes et al., 2013*).

We next investigated the extent of spontaneous oligodendrocyte differentiation from these progenitors in culture, since a substantial subset of broad NG2+ progenitors produces oligodendrocytes *in vivo* (*Zhu et al., 2008a*). Previous work demonstrated that *Sox6* is expressed by at least some proliferating NG2+ progenitors, and is down-regulated upon differentiation (*Baroti et al., 2016*; *Stolt et al., 2006*). Under our culture conditions, FACS-purified cortical SOX6+/NG2+ progenitors continue to express *Sox6* (*Figure 1F, I, J*), indicating maintenance of their progenitor state. ICC for O4 expression (a marker for pre-myelinating oligodendrocytes) revealed that only ~0.15% of these cells express O4 at 3 and 5 DIV (~51 and~49 O4+ cells/cm$^2$, respectively). Similarly, qPCR for myelin basic protein (*Mbp*), a canonical oligodendrocyte marker, demonstrated that *Mbp* expression does not increase when cells are cultured for 3 or 5 days, compared to acutely sorted progenitors (n=4) (*Figure 1—figure supplement 3E*). Together, these data indicate that our culture conditions are not permissive for oligodendrocyte differentiation, and that the purified SOX6+/NG2+ progenitors maintain their progenitor state.

Next, we applied multiple analyses to identify whether there exist contaminant neurons or astrocytes in these cultures of SOX6+/NG2+ progenitors. To identify non-progenitor cells in culture, we immunolabeled for TUJ1 (antibody against TUBB3, a common immature neuronal marker) and GFAP (expressed by astrocytes and some other types of neural progenitors) at 3, 5, and 7 DIV (*Figure 1K*). At 3 DIV, among ~12,000 total cells/cm$^2$, there were 7 TUJ1+ cells and 3 GFAP+ cells. At 5 DIV, among ~32,000 total cells/cm$^2$, there were 11 TUJ1+ cells and 0 GFAP+ cells. At 7 DIV, among ~70,000 total cells/cm$^2$, there were 6 TUJ1+ cells and 14 GFAP+ cells (*Figure 1K*). These data reveal the exceptional purity (>99.9% pure) of these primary cultures of SOX6+/NG2+ cortical progenitors. Reinforcing these immunocytochemical results, qPCR revealed that neither *Tubb3* nor *Gfap* are detected in these cultures at 5 DIV, nor in acutely sorted DsRed-Bright cells (n=4) (*Figure 1—figure supplement 3E*). In striking contrast, and reinforcing that these culture conditions maintain progenitor competence of SOX6+/NG2+ progenitors, supplementing medium with serum resulted in downregulation of *Sox6* and *NG2* and increased expression of *Gfap* (n=4) (*Figure 1—figure supplement 3Q*). Together, these results identify that there is essentially no contamination under these culture conditions at any time point investigated, and that progenitors maintain their molecular and functional characteristics *in vitro*.

We further investigated the progenitor cultures for potential pericyte contamination, since pericytes express NG2 proteoglycan (*Ozerdem et al., 2001*), so they are DsRed-positive in NG2-DsRed cortex (*Figure 1D*, *Figure 1—figure supplement 3A and B*). qPCR for pericyte markers *Pdgfrb* and *Mcam (CD146)* revealed that pericytes are abundant in acutely sorted DsRed-Bright cultures, but are absent in culture at 5 DIV (n=4) (*Figure 1—figure supplement 4F*), indicating that pericytes do not survive in these culture conditions. Validating these results by ICC, there were no PDGFRB+ cells in culture at either 3 or 5 DIV (0 cells/cm$^2$, n=2), unless DsRed-Bright cells were cultured in serum-supplemented media (*Figure 1—figure supplement 4J–L*). Together, these results reveal that these culture conditions do not support pericyte survival, and that progenitor cultures are pericyte-free.

To even further investigate by independent means whether progenitors maintain their *in vivo* molecular features *in vitro*, we performed RNA-seq on these cultures at 5 DIV (n=6), evaluating expression of 500 genes most enriched in the major alternative cell lineages (*Supplementary file 1*; *Zhang et al., 2014*). The purified SOX6+/NG2+ progenitor cultures express progenitor-enriched genes (*Figure 1L*), but, appropriately, do not express neuronal-, astroglial-, microglial-, pericyte-, or vascular-enriched genes (*Figure 1M, N*, *Figure 1—figure supplement 4A–D and G–I*), confirming the ICC and qPCR results. Similarly, oligodendrocyte-enriched genes are not upregulated in culture compared to acutely sorted cells (*Figure 1M, N*, *Figure 1—figure supplement 4E*). Importantly, gene expression profiles of cultured progenitors were highly consistent and reproducible across biological replicates (n=6) (*Figure 1—figure supplement 4A–I*). Together, these data further confirm that cortical SOX6+/NG2+ progenitors maintain their molecular characteristics *in vitro*, enabling establishment of a robust *in vitro* culture system in which to reproducibly manipulate progenitors under controlled conditions.

## Multi-gene construct 'NVOF' induces neuronal differentiation and unipolar pyramidal morphology from SOX6+/NG2+ cortical progenitors

To direct differentiation of corticospinal neurons from cortical SOX6+/NG2+ progenitors, we designed a tandem construct containing three transcriptional controls (**N**eurog2, **V**P16:**O**lig2, and **F**ezf2 – collectively termed '**NVOF**') based on their developmental functions (*Figure 2A and B*; *Tang et al., 2009*). The expression of the polycistronic construct is driven by the CMV-β-actin (CAG) promoter, with the open reading frames separated by 2A linker sequences (*Supplementary file 3*; *Tang et al., 2009*), also including a GFP reporter to identify transfected cells.

First, to drive glutamatergic neuronal identity, we selected the pallial proneural transcription factor neurogenin2 (*Neurog2*) (*Schuurmans et al., 2004*; *Mattar et al., 2008*). Previous data showed that forced expression of *Neurog2* reprograms cultured postnatal glia and human ESC/iPSCs into synapse-forming glutamatergic neurons *in vitro* (*Heinrich et al., 2010*; *Zhang et al., 2013*; *Hulme et al., 2022*), and can induce neuron-like cells from postnatal glial cells (*Felske et al., 2023*; *Herrero-Navarro et al., 2021*) and injury-induced reactive glial cells in the adult mouse brain (*Gascón et al., 2016*). We tested *Neurog2* alone in cultured progenitors and found that, in line with previous reports, *Neurog2* is sufficient to induce neurons with long axons *in vitro* (*Figure 2—figure supplement 1A*).

Second, to overcome the predominant gliogenic potential in NG2+ progenitors, we complemented *Neurog2* with *VP16:Olig2* (VP16 transactivation domain from herpes simplex virus fused to an OLIG2 DNA binding domain) (*Mizuguchi et al., 2001*). This activator form of *Olig2* functions as a dominant negative transcriptional regulator to counteract *Olig2* gliogenic function (*Mizuguchi et al., 2001*; *Novitch et al., 2001*; *Zhou et al., 2001*). *Olig2*, a bHLH transcription factor, is necessary for the specification of a broad population of NG2+ progenitors and for their differentiation into oligodendrocytes (*Li and Richardson, 2016*). In addition, OLIG2 has been shown to antagonize NEUROG2 activity during neurogenesis to maintain progenitors for subsequent gliogenesis during spinal cord development (*Lee et al., 2005*). Misexpression of *Olig2* in the cortex broadly represses proneural and neurogenic genes and increases oligodendrocyte precursor cell numbers (*Liu et al., 2015*). Intriguingly, antagonizing OLIG2 function in reactive glial cells after injury results in a substantial number of immature neurons in the cortical or striatal parenchyma (*Buffo et al., 2005*; *Kronenberg et al., 2010*). To confirm whether *VP16:Olig2* is able to suppress glial differentiation capacity of cortical SOX6+/NG2+ progenitors in our experimental paradigm, we transfected progenitors with either *VP16:Olig2* or control GFP constructs. At 1 DPT, the cultures were treated with thyroid hormone (T3) to induce differentiation of oligodendrocytes. At three days post-T3 treatment, as expected, control

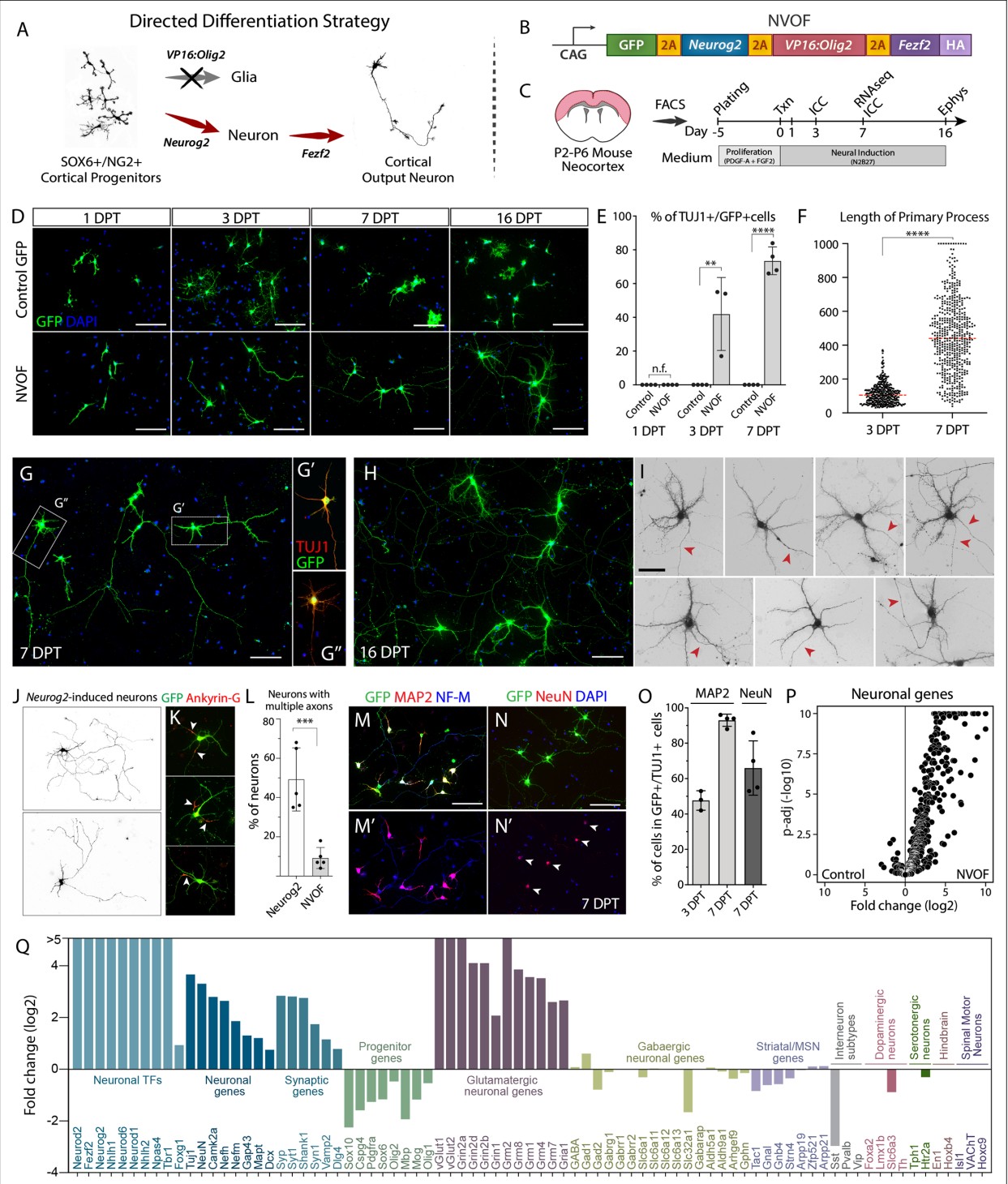

**Figure 2.** NVOF induces mature glutamatergic neurons from SOX6+/NG2+ cortical progenitors *in vitro*. (**A–C**) Strategy for directed differentiation of SOX6+/NG2+ progenitors into cortical output neurons (**A**), the NVOF multigene construct (**B**), and the experimental outline (**C**). (**D**) Representative images of control-GFP and NVOF-transfected cells at 1-, 3-, 7-, and 16-days post-transfection (DPT). Unlike control-transfected cells, NVOF-transfected cells lose progenitor morphology at 1 DPT and progressively exhibit complex neuronal morphology, including a primary axon-like process and multiple dendrite–like processes. (**E**) Percentage of control-GFP and NVOF-transfected cells with neuronal morphology and TUJ1 expression (~42% at 3 DPT and ~74% at 7 DPT for NVOF, n=4, >200 cells/n). (**F**) Quantification of primary process length for NVOF-induced neurons at 3 and 7 DPT (n=3,>100 cells/n). (**G**) Representative morphology of NVOF-induced, TUJ1+ neurons at 7 DPT. Note the single axon, dendrite-like structures, and multiple axonal collaterals. (**H**) Representative images of NVOF-induced neurons at 16 DPT showing acquisition of elaborate dendritic morphology and highly intercalated axonal processes. (**I**) High-power representative images of individual NVOF-induced neurons at 16 DPT showing dendritic complexity and

*Figure 2 continued on next page*

*Figure 2 continued*

a single primary axon-like process for each neuron (red arrows). (**J**) Representative images of *Neurog2*-induced neurons with multiple atypical axon-like processes. (GFP is pseudocolored for enhanced clarity of cell morphology). (**K**) Representative images of *Neurog2*-induced neurons expressing the axonal marker ANKYRIN-G (ANK3) by multiple neurites (n=2). (**L**) Quantification of neurons with single versus multiple axons in *Neurog2*- and NVOF-induced neurons. At 7 DPT, 49 ± 16% of *Neurog2*-induced neurons have multiple, long axon-like processes, whereas a small number of such neurons exist after NVOF induction (9 ± 5%) (n=5, >100 cell). See methods for details. (**M–N**) Representative images of NVOF-induced neurons at 7 DPT showing compartmentalized expression of the somato-dendritic marker MAP2, the somato-axonal marker Neurofilament-M, and the mature neuronal marker, NeuN. (**O**) Quantification of NVOF-induced, TUJ1+ neurons expressing MAP2 at 3 DPT (~48%, n=3, >200 cells) and 7 DPT (~93%, n=4, >200 cells), as well as NeuN at 7 DPT (66 ± 16%, n=4, >100 cells). (**P**) Volcano plot showing upregulation of neuronal genes in NVOF-induced neurons compared to control-transfected cells at 7 DPT (RNA-seq, n=3, biological replicates). (**Q**) Bar graph of RNA-seq data displaying upregulation of neuronal genes and downregulation of progenitor genes in NVOF-induced neurons at 7 DPT. Neurons exclusively upregulate glutamatergic genes, but not genes specific to alternate neuronal identities. Scale bars (**D, G, H, J, M, N**) 100 µm; (**I**) 50 µm. Error bars show standard deviations. ****$p<0.0001$, ***$p<0.001$, **$p<0.01$, t-test in (**E, F, L**). n.f. (no TUJ1+ cell found).

The online version of this article includes the following figure supplement(s) for figure 2:

**Figure supplement 1.** Expression and function of individual NVOF components.

**Figure supplement 2.** NVOF redirects the axons of later-born upper layer neurons to subcortical targets, similar to deep layer cortical output neurons.

**Figure supplement 3.** NVOF-transduced progenitors rapidly lose progenitor identity and acquire cardinal features of mature functional neurons.

**Figure supplement 4.** NVOF induces glutamatergic neurons from SOX6+/NG2+ progenitors with high fidelity and reproducibility.

**Figure supplement 5.** Synthetic mRNA mediates neuronal induction from SOX6+/NG2+ cortical progenitors.

cells differentiated into oligodendrocyte-like cells, whereas *VP16:Olig2* transfected progenitors had remarkably turned into neuroblast-like bipolar cells, indicating that VP16:OLIG2 successfully blocks endogenous OLIG2 function (*Figure 2—figure supplement 1B and C*).

Third, to induce cortical output neuronal fate, we selected *Fezf2*, an upstream transcriptional regulator that controls specification and development of cortical output neurons during cortical neurogenesis (*Greig et al., 2013*; *Chen et al., 2004*; *Arlotta et al., 2005*; *Galazo et al., 2023*; *Figure 2—figure supplement 1D and E*, see also Discussion). *Fezf2* is capable via single gene over-expression of generating cortical output neuronal fate from alternate cortical progenitors (*Molyneaux et al., 2005*), from progenitors of striatal neurons *in vivo* (*Rouaux and Arlotta, 2010*), and from intracortical projection neurons post-mitotically in the early postnatal brain (*De la Rossa et al., 2013*).

We first verified expression of individual proteins from the polycistronic construct (*Figure 2—figure supplement 1H–J*), then assessed the construct's functionality in mouse embryonic cortical progenitors *in vivo* (*Figure 2—figure supplement 2*). Previous work has shown that misexpression of *Fezf2* in late-stage embryonic cortical progenitors modifies their fate to cortical output neurons, re-routing the intracortical axonal trajectories of layer II/III neurons to subcortical targets (*Molyneaux et al., 2005*). To investigate whether this FEZF2 function persists in the presence of NEUROG2 and VP16:OLIG2, we electroporated NVOF into embryonic ventricular zone progenitors *in utero* at E15.5, the peak production of upper layer intracortical neurons, and found that forced expression of NVOF induces cortical output identity in electroporated neurons (n=3) (*Figure 2—figure supplement 2*). Unlike control GFP-only neurons (*Figure 2—figure supplement 2A and B*), many NVOF+ axons descend through the internal capsule, to or past the thalamus (*Figure 2—figure supplement 2C*), with some extending into the cerebral peduncle (*Figure 2—figure supplement 2D*). These data demonstrate that the NVOF construct is expressed by electroporated neurons, and that *Fezf2* continues to specify cortical output identity when co-expressed with *Neurog2* and *VP16:Olig2*.

We transfected NVOF into cultured cortical SOX6+/NG2+ progenitors at 4–5 days after FACS purification and analyzed their morphology and expression of cardinal ICC markers of cell type identity over two weeks of differentiation (*Figure 2C*). Progenitors began to lose multipolar morphology within 24 hr (*Figure 2D*). By 3 days post-transfection (DPT), many extended a single axon-like neurite (*Figure 2D and F*) and expressed the broad neuronal marker TUJ1 (42%, n=3, >200 cells/experiment) (*Figure 2E*, *Figure 2—figure supplement 3A and B*). This morphological transformation was coupled with the loss of the progenitor markers NG2 and SOX10 (*Figure 2—figure supplement 3A and B*). By 7 DPT, ~73% of NVOF-transfected cells expressed TUJ1, acquired neuronal morphology with dendrite-like features, and extended a single prominent axon-like process (n=4, >200 cells/experiment) (*Figure 2D-G*, *Figure 2—figure supplement 3C and D*). Consistent with pyramidal neuron morphology, the primary axon-like processes of NVOF-directed neurons underwent significant

extension between 3 DPT and 7 DPT, often extending further than 500 μm from the soma (>40%, n=3) (*Figure 2F*). By 16 DPT, the morphology of these putative neurons became more elaborate; the single long axon-like neurite was maintained, their dendrite-like structures became more tufted, and axon-neurite branches of neighboring cells became intercalated (*Figure 2H and I*).

In striking contrast, progenitors transfected with a control GFP-only construct displayed glial morphology throughout the culturing period, and no GFP+/TUJ1+ cells were present at all (n=4, 250–350 cells/experiment) (*Figure 2D and E*). Furthermore, even among non-transfected, GFP-negative cells, only 5 cells/cm$^2$ out of ~30,000 progenitors/cm$^2$ were TUJ1+, and these GFP-/TUJ1+ cells did not increase over time (n=4). These results further reinforce the absence of contaminating progenitors with spontaneous neurogenic characteristics in these cultures, and the lack of spontaneous differentiation by cultured cortical SOX6+/NG2+ progenitors.

*Neurog2* is widely used to induce generic excitatory neurons from somatic and pluripotent stem cells (*Heinrich et al., 2010*; *Zhang et al., 2013*). We directly compared *Neurog2*-induced and NVOF-induced neurons to determine whether *Neurog2* might be sufficient for induction of equivalent neuronal differentiation from cultured SOX6+/NG2+ cortical progenitors. We transfected cultured progenitors with either *Neurog2*-GFP or NVOF and analyzed cells at 7 DPT. Though superficially similar in some respects to NVOF-induced neurons (*Figure 2J*), *Neurog2* induces multipolar neuronal morphology with many dendrite-like structures and multiple long axon-like processes. While almost all NVOF-induced neurons extend a single primary axon (90%), ~50% of *Neurog2*-induced neurons aberrantly extend multiple axon-like ANKYRIN-G+ processes originating from their cell bodies (*Figure 2J–L*) (n=5, >100 cells/n). This aberrant, over-exuberant neuritogenesis by *Neurog2*-induced neurons indicates defective polarization, potentially due to a lack of negative feedback signaling for inhibition of surplus axon formation (*Funahashi et al., 2020*).

## NVOF-induced neurons exhibit cardinal features of mature functional neurons

We investigated further whether NVOF-induced, TUJ1+ cells acquire the cardinal molecular hallmarks of mature neurons. At 7 DPT, NVOF-induced neurons express the somato-dendritic marker MAP2 (>90%, n=4, 130–200 cells/n) and the somato-axonal marker NF-M (*Figure 2M and O*), indicating clear polarization and dendritic compartmentalization. Dendrite formation was confirmed by high-power imaging at 16 DPT, revealing that the NVOF-induced neurons have dendrite-like processes with filopodial protrusions and a single axon-like primary process lacking dendrite-like structures (*Figure 2I*, highlighted with red arrows). Further, at 7 DPT, NVOF-induced neurons express neuronal nuclear antigen (NeuN) (66 ± 16%, n=4, >100 cells/n) (*Figure 2N and O*), polysialylated neural cell adhesion molecule (PSA-NCAM or *Ncam1*) (*Figure 2—figure supplement 3F and G*), the presynaptic molecule synapsin (*Figure 2—figure supplement 3H*), with some displaying synaptophysin in axonal branches and tips of axonal protrusions (*Figure 2—figure supplement 3I*), and vGLUT1 (vesicular glutamate transporter 1) (*Figure 2—figure supplement 3J*), indicating glutamatergic identity. Together, these data indicate that NVOF robustly induces neuronal differentiation and maturation by cortical SOX6+/NG2+ progenitors *in vitro*.

To determine whether neuronal differentiation from these cortical SOX6+/NG2+ progenitors requires an intermediate proliferative step, we pulsed cultures with BrdU for 15 hr after transfection and labeled GFP+ cells (NVOF-transfected or GFP-only controls) for BrdU by ICC at 3 DPT (n=2). Previous work has reported that cell division is not required for neuronal differentiation from resident glia (*Heinrich et al., 2010*). While a majority of cells transfected with GFP-only were BrdU+, only rare NVOF-transfected cells were BrdU+. This result indicates that NVOF causes rapid cell cycle exit, and that chromatin reorganization during cell division is not required for NVOF-induced neuronal differentiation and maturation from SOX6+/NG2+ progenitors.

To more broadly investigate the molecular identity and specificity of neurons induced from cortical SOX6+/NG2+ progenitors transfected with NVOF, we performed RNA-seq on control GFP-transfected and NVOF-transfected cells at 7 DPT (n=3) (*Figure 2—figure supplement 4A and B*). NVOF-induced neurons have decreased expression of progenitor genes and increased expression of neuronal genes, relative to GFP-transfected cells (*Figure 2M*, *Figure 2—figure supplement 4C*). Upregulated neuronal genes include proneural transcription factors, neuron-specific cytoskeletal molecules, and molecules that function in synaptic transmission, dendritic specialization, glutamatergic signaling,

axon guidance, and neuronal connectivity (*Figure 2P, Q*, *Figure 2—figure supplement 4D*). *Neurog2* is required for the differentiation of multiple neuronal types across regions of the nervous system and overexpression of *Neurog2* in somatic and stem cells generates neurons with mixed identities (*Hulme et al., 2022*; *Lin et al., 2021*; *Kempf et al., 2021*; *Chouchane et al., 2017*; *Sheta et al., 2022*; *Ang et al., 2024*). We, therefore, confirmed that NVOF-induced neurons express exclusively genes typical of glutamatergic neurons, but not genes specific for alternate neuronal types (e.g. GABAergic interneurons, striatal projection neurons, or serotonergic, dopaminergic, hindbrain, or spinal motor neurons) (*Figure 2Q*, *Figure 2—figure supplement 4E and N*).

We co-cultured NVOF-transfected cells at 1 DPT with primary forebrain cells from mouse cortex in astrocyte-conditioned media (*Figure 2—figure supplement 3K*) (see Methods) to investigate whether such a potentially permissive and/or instructive environment might even further enhance neuronal differentiation and maturation. It is known that neurons cultured below critical density, or in the absence of glial-derived trophic factors, often survive poorly and/or do not mature (*Kaech and Banker, 2006*; *Pfrieger and Barres, 1997*). Indeed, culture with primary neurons increased morphological maturation of NVOF-induced neurons, resulting in elaborate dendrites with abundant synapses (n=2) (*Figure 3A–D*, *Figure 2—figure supplement 3L–N*), demonstrating synaptic input from surrounding neurons and functional integration into neuronal networks. Quite notably, the morphology and density of dendritic synapse-like structures in NVOF-induced neurons were essentially indistinguishable from those of primary cortical neurons cultured under identical conditions (*Figure 3*).

To investigate functional properties of NVOF-induced neurons, we performed whole-cell patch-clamp recordings at 10 DPT (without co-culture) and at 16 DPT (with primary neuron co-culture) (*Figure 3E–P*). Consistent with their immunocytochemical and morphological characteristics, NVOF-induced neurons possess electrophysiological hallmarks of neurons, including trains of action potentials upon depolarizing steps (*Figure 3E–G*), HCN-channel currents (Isag) upon hyperpolarization (*Figure 3H*), and spontaneous synaptic currents (*Figure 3O and P*). NVOF-induced neurons also mature over time in culture, with overall increases in the action potential threshold (–35.9 mV at 10 DPT versus –30.9 mV at 16 DPT) (*Figure 3K*), decreases in action potential width (2.1 ms at 10 DPT versus 1.3 ms at 16 DPT) (*Figure 3M*), and increases in Isag (3.4 mV at 10 DPT versus 12.1 mV at 16 DPT) (*Figure 3N*). Cortical SOX6+/NG2+ progenitors transfected with the control vector possess membrane resistances and resting voltages that are inconsistent with neuronal identity (*Figure 3I and J*).

## Vector-free induction of neuronal differentiation from cortical SOX6+/NG2+ progenitors with synthetic modified mRNAs

The results presented above reveal that NVOF-induced neurons express a quite comprehensive set of molecules that indicate faithful neuronal differentiation, and that they possess electrophysiological properties indistinguishable from those of primary neurons. However, previous work reports that sustained expression of *Neurog2* can be deleterious to differentiating cortical neurons (*Cai et al., 2000*). To more closely reproduce the dynamics of developmental expression of *Neurog2*, we aimed to restrict *Neurog2* expression to a short, early time period using synthetic, chemically-modified RNA in which one or more nucleotides are replaced by modified nucleotides. Previous work, in multiple systems, has revealed that synthetic modified mRNA mediates highly efficient, integration-free, transient protein expression *in vitro* and *in vivo* without eliciting an innate immune response (*Sahin et al., 2014*; *Warren et al., 2010*).

In contrast to the transient expression of *Neurog2* during neurogenesis *in vivo*, cortical output neurons express *Fezf2* throughout development and adulthood (*Molyneaux et al., 2005*). To emulate the distinct kinetics of endogenous developmental expression of *Neurog2* and *Fezf2*, we devised a strategy by which *Neurog2* is transiently expressed via synthetic modified mRNA, and *Fezf2* is expressed on an ongoing basis as a plasmid DNA construct with a constitutively active CAG promoter. We first adapted our transfection protocol to transfect cortical SOX6+/NG2+ progenitors with mRNA at high efficiency (*Figure 2—figure supplement 5A and B*). To test the feasibility of DNA-RNA co-transfection, we co-transfected tdTomato as a plasmid DNA, and GFP as a synthetic modified mRNA (*Figure 2—figure supplement 5C*). ~50% of fluorescent cells were co-transfected with both reporters (n=3). We investigated the dynamics of protein expression, finding that the GFP synthetic

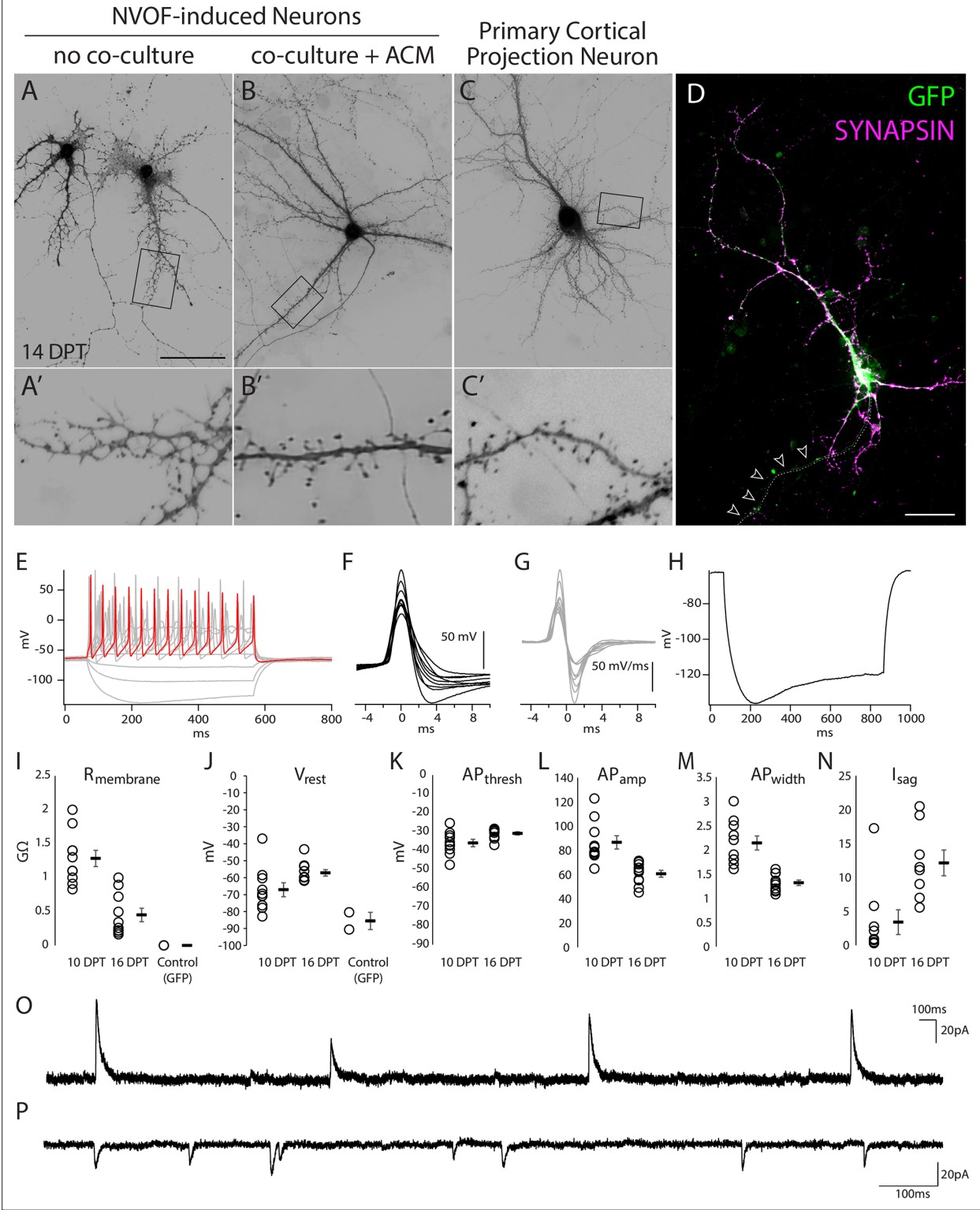

**Figure 3.** NVOF-induced neurons are electrically active and have spontaneous synaptic currents. (**A–B**) Representative high-magnification images of NVOF-induced neurons at 14 DPT (pseudo-colored GFP) with and without coculture of forebrain primary neurons and astrocyte-conditioned media. (**A′-B′**) Insets showing the boxed areas. Note differences in morphology of presumptive synaptic structures between the two conditions. See methods for details. (**C**) Representative high-magnification image of a primary cortical neuron at 14 DPT (pseudo-colored tdTomato) from *in utero* electroporated wild-type mice as a positive control. (**C′**) Note similarity in morphology of presumptive synaptic structures between primary neurons and NVOF-induced neurons in B′. See methods for details. (**D**) Representative high-magnification image of a SYNAPSIN-positive NVOF-induced neuron co-cultured with

*Figure 3 continued on next page*

*Figure 3 continued*

forebrain neurons, indicating abundant connections from surrounding neurons. Arrows show the presumptive single primary axon with no synapsin staining. (**E**) A representative NVOF-induced neuron at 10 DPT showing depolarizing steps evoking a train of action potentials (red highlighted trace: step 6, 50 pA). 10 min after break-in, or following a resting Vm stabilization greater than 1 min, cells were injected with 10 current steps ranging from –40 pA to 95 pA in 15 pA increments, for a duration of 500 ms each. (**F**) The first evoked action potential in response to positive current injections for 10 individual cells, overlaid (10 DPT). Waveforms are aligned at threshold for comparison. (**G**) Corresponding dV/dt traces for action potentials shown in F. (**H**) Representative sag current, indicating presence of Ih, induced with a 500 ms current injection of –40 pA (average of 10 sweeps). (**I**) Cell membrane resistance decreases over time (10 DPT, n=10; 16 DPT, n=10), and is substantially lower without NVOF (GFP, n=2). (**J**) Resting membrane voltage for each condition (10 DPT, n=10; 16 DPT, n=10; GFP, n=2). (**K–M**) Action potential threshold, amplitude, and width at 10 DPT (n=10) and 16 DPT (n=10). (**N**) Sag current at 10 DPT (n=9) and 16 DPT (n=8). (**O**) Representative spontaneous outward synaptic currents recorded at –70 mV in NVOF+ cells at 16 DPT. (**P**) Representative spontaneous inward synaptic currents recorded at –70 mV in NVOF-induced neurons at 16 DPT. Scale bars (**A–D**) 50 μm; (**F**) 50 mV; (**G**) 50 mV/ms. For all graphs I-N, open circles are individual cells, filled boxes are mean (±) s.e.m.

modified mRNA displays peak protein levels 12–24 hr post-transfection, then declines (*Figure 2—figure supplement 5D–G*).

Next, we directly compared the efficacies of a *Neurog2* DNA construct and a *Neurog2* synthetic modified mRNA. Strikingly, confirming the neurogenic competency of cortical SOX6+/NG2+ progenitors, one dose of *Neurog2* synthetic modified mRNA induces robust neurogenesis, albeit with lower efficiency than *Neurog2* DNA or NVOF (n=3) (*Figure 2—figure supplement 5H and I*). We then co-expressed *Neurog2* in synthetic modified mRNA form and *Fezf2* as a plasmid DNA construct. Remarkably, this combination of synthetic modified mRNA plus plasmid DNA produced abundant neurons morphologically indistinguishable from NVOF-induced neurons (*Figure 2—figure supplement 5J*). These results reveal that synthetic modified mRNA transfection can be used to tailor more precise kinetics of developmental genes toward directed differentiation of neuronal subtypes.

## NVOF-induced neurons acquire molecular hallmarks of cortical output neuron identity *in vitro*

We progressively focused our investigations to evaluate whether NVOF-induced neurons *in vitro* express cardinal molecular hallmarks of endogenous cortical output neurons, with a particular focus on the major output neuron subgroup of subcerebral projection neurons (SCPN, comprising neuronal subtypes that project to brainstem and spinal cord). Results reveal that ~58% of NVOF-induced neurons at 7 DPT express BCL11b/CTIP2, a transcription factor that regulates outgrowth, guidance, and fasciculation of SCPN/CSN axons (*Arlotta et al., 2005*) (n=6, ave 177 cells/experiment) (*Figure 4A, G and H*), whereas no control GFP-only cells express CTIP2 (n=2, ave 207 cells/experiment). NVOF-induced neurons also express PCP4 (Purkinje cell protein 4), a calmodulin-binding protein reproducibly expressed by SCPN/CSN (*Arlotta et al., 2005*) (~83% at 7 DPT, n=4, ave 131 cells/experiment) (*Figure 4B, G and H*). Importantly, the number of CTIP2+ NVOF-induced neurons continued to increase over time, indicating continued subtype differentiation after 7 DPT (*Figure 4I*).

Next, we investigated NVOF-induced neurons for expression of corticothalamic projection (CThPN) neuron-enriched molecular controls. Intriguingly, most NVOF-induced neurons express FOG2 (ZFPM2) (~79% at 7 DPT, n=4, ave 132 cells/experiment) (*Figure 4C, G and H*), a critical regulator of CThPN axonal targeting and diversity (*Galazo et al., 2016*). However, FOXP2, a transcriptional control required for CThPN specification (*Hisaoka et al., 2010*), is expressed heterogeneously by NVOF-induced neurons, with minimal to no expression by many neurons (*Figure 4D*). These data indicate that NVOF-induced neurons acquire broad cortical output neuronal identity, but refinement of subtype identity (SCPN vs. CThPN) is incomplete, suggesting that additional controls are required for complete subtype refinement.

We also investigated the possibility of subtype 'confusion' during directed differentiation by examining whether NVOF-induced neurons also or alternatively express cardinal molecular markers of callosal projection neurons (CPN) or other intra-cortical projection neurons. If identified, this would indicate either immature differentiation or mixed/hybrid identity that is commonly observed with ES/iPSC-derived neurons (*Sadegh and Macklis, 2014*). Quite notably and appropriately, NVOF-induced neurons do not express SATB2 (0% at 7 DPT, n=4, ~130 cells/n) (*Figure 4E and G*) or CUX1 (n=3) (*Figure 4F*), molecular controls that are expressed by CPN and other intracortical projection neurons.

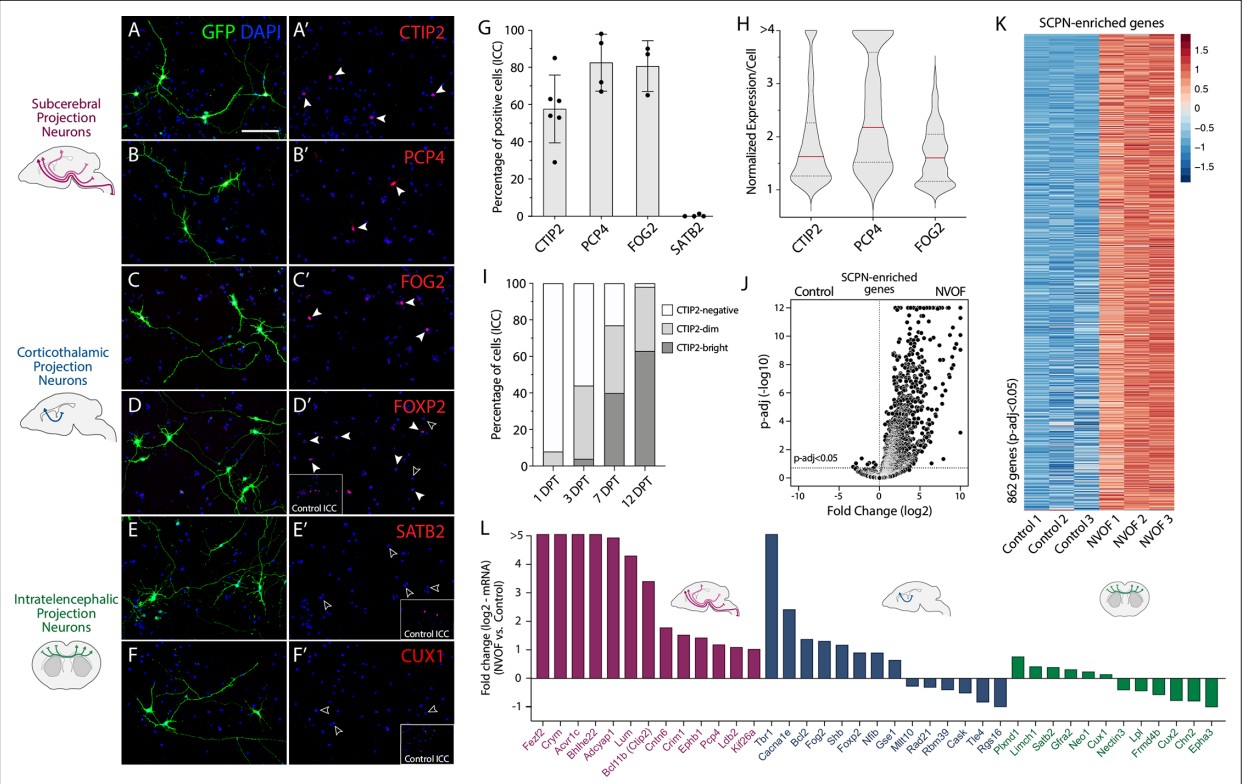

**Figure 4.** NVOF-induced neurons exhibit molecular hallmarks of corticospinal neurons *in vitro*. (**A–F**) Representative immunocytochemistry images of NVOF-induced neurons expressing the subcerebral projection neuron (SCPN) transcriptional controls CTIP2 (**A**) and PCP4 (**B**), the corticothalamic projection neuron (CThPN) transcriptional controls FOG2 (**C**) and FOXP2 (**D**), but not the callosal projection neuron (CPN) molecular controls SATB2 (**E**) and CUX1 (**F**). Scale bars (**A–F**) 100 μm. (**G**) Percentage of NVOF-induced, TUJ1+ neurons expressing CTIP2 (56 ± 20%, n=5), PCP4 (77 ± 14%, n=3), FOG2 (81 ± 13%, n=3), and SATB2 (0%, n=3). Error bars show standard deviations. (**H**) Violin plot shows mean intensities of CTIP2, PCP4, and FOG2 fluorescence signals in nuclei of NVOF-induced neurons. Plotted values are mean nuclear intensity of individual neurons normalized to the average intensity of the three lowest-expressing neurons. Red line shows median expression, and dark gray lines show quartile expressions. (**I**) Bar plot showing percentages of CTIP2-negative, -dim, and -bright neurons at 1-, 3-, 7-, and 12-DPT (n=1). (**J, K**) Volcano plots and heatmaps of neurons transfected with control GFP and NVOF 7 DPT, displaying the 862 genes enriched in SCPN primary neurons compared to control-transfected cells. See methods for details. (**L**) Bar plot of RNA-seq data showing upregulation of SCPN (purple) and CThPN (blue) marker genes, and no activation or downregulation of CPN (green) genes by NVOF-induced neurons relative to neurons transfected with control GFP at 7 DPT.

Reinforcing and extending these ICC results, RNA-seq reveals that NVOF-induced neurons express many SCPN/CSN-enriched genes (***Figure 4K***) (see methods), including key molecules with central functions in subtype specification of SCPN/CSN (***Figure 4L***), along with some CThPN-enriched genes (e.g. *Tbr1*, *Fog2*, and *Foxp2*) (***Figure 4L***). In accordance with the ICC results, RNA-seq reveals that NVOF-induced neurons have no or minimal expression of genes specific to CPN or other intracortical projection neurons, including *Satb2*, *Cux1*, and *Cux2* (***Figure 4L***). Together, these results indicate that NVOF-induced neurons acquire cortical output neuron identity, primarily of SCPN/CSN, but with some CThPN features, without fully refining molecular identity between these subtypes of cortical output neurons (see Discussion).

We directly compared expression of key subtype-specific molecular controls between NVOF-induced and *Neurog2*-induced neurons (***Figure 5—figure supplement 1***). While *Neurog2*-induced neurons approximate elements of NVOF induction, with some expression of cortical output neuron markers CTIP2, PCP4, and FOG2 (***Figure 5—figure supplement 1A***), and not the CPN and other intracortical neuronal molecules, such as SATB2 (***Figure 5—figure supplement 1A***) and CUX1 (data not shown) by ICC, NVOF induction generated more neurons expressing CTIP2, PCP4, and FOG2, with higher average expression (n=>3 for each marker) (***Figure 5—figure supplement 1A and B***), indicating substantially enhanced subtype-specific differentiation by *Fezf2*. Reinforcing the interpretation from aberrant multipolar morphology of *Neurog2*-induced neurons that *Neurog2* alone induces

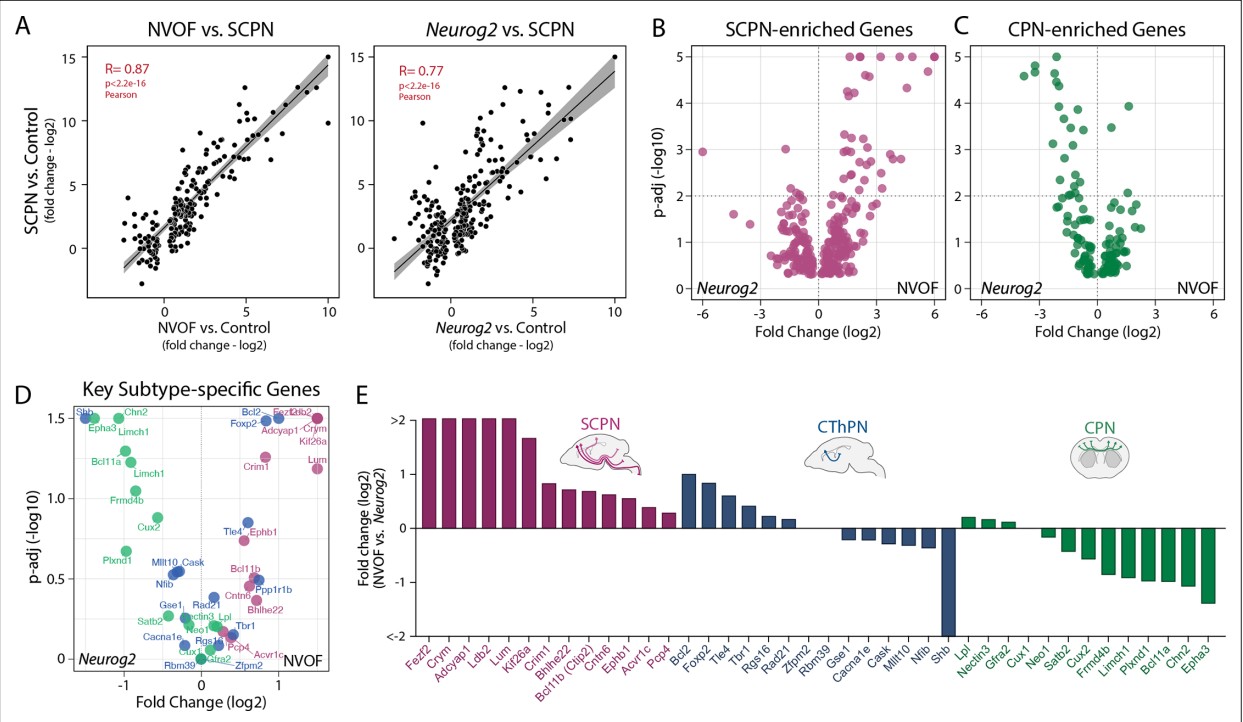

**Figure 5.** Unlike NVOF-induced neurons, *Neurog2*-induced neurons exhibit unresolved subtype-specific molecular features. (**A**) Pearson correlation analysis shows high similarity between NVOF-induced neurons at 7 DPT and primary subcerebral projection neuron (SCPN) at P2 (R: 0.87). Compared to NVOF, *Neurog2* induction (7 DPT) leads to decreased similarity with primary SCPN at P2 (R: 0.77). Data points are log2 fold differences of gene expression at 7 DPT by NVOF- or *Neurog2*-induced neurons (on X-axis) and by SCPN (on Y-axis) compared to progenitors transfected with control GFP. (**B**) Volcano plot showing fold differences of SCPN-enriched genes between NVOF- and *Neurog2*-induced neurons. (**C**) Volcano plot showing fold differences of CPN-enriched genes between NVOF- and *Neurog2*-induced neurons. (**D, E**) Direct comparison of NVOF- versus *Neurog2*-induced neurons at 7 DPT for select developmental genes with key roles in specification and differentiation of SCPN, CPN, and CThPN. Scatter plot (**D**) and bar graph (**E**) shows fold differences in gene expression.

The online version of this article includes the following figure supplement(s) for figure 5:

**Figure supplement 1.** *Neurog2* is not sufficient to induce molecular hallmarks of cortical output neurons.

'confused' and unresolved differentiation (*Figure 2J–L*), *Neurog2*-induced neurons simultaneously express CTIP1 (BCL11a), a CPN molecular control and antagonist of CTIP2 (*Greig et al., 2016*; *Woodworth et al., 2016*; *Figure 5—figure supplement 1A–D*). During cortical development, CTIP1 is initially expressed broadly by postmitotic neurons, but later, through its cross-repressive interaction with CTIP2, its expression resolves to CPN and CThPN, but not SCPN/CSN, at E17. Continued expression of CTIP1 by *Neurog2*-induced neurons further indicates incomplete and unresolved subtype differentiation.

To comprehensively and directly characterize subtype identities induced by NVOF compared with FACS-purified primary SCPN/CSN or the morphologically and molecularly 'hybrid' *Neurog2*-induced neurons, we performed RNA-seq on FACS-purified GFP+ neurons generated by NVOF or *Neurog2* at 7 DPT (n=3), and on FACS-purified SCPN/CSN or CPN from P2 mice (n=3) (*Figure 5*). Pearson correlation analysis for genes enriched in SCPN compared to CPN reveals that NVOF-induced neurons are substantially more similar to primary SCPN/CSN (R=0.87) than are *Neurog2*-induced neurons (R=0.77) (*Figure 5A*). Even more strikingly, NVOF induces higher expression of many SCPN/CSN genes relative to *Neurog2* alone (*Figure 5B*), while *Neurog2* simultaneously and aberrantly activates many typically CPN-specific genes that are expressed at E15 in mouse, the peak period of CPN birth and specification (*Figure 5C*; *Molyneaux et al., 2015*). In particular, in line with the prior ICC results, NVOF-induced neurons express SCPN/CSN genes with known key functions in subtype-specific development of SCPN/CSN at higher levels (e.g. *Ctip2* and *Ephb1*, both essential for SCPN/CSN axon guidance) (*Figure 5D and E*; *Arlotta et al., 2005*; *Lodato et al., 2014*). NVOF-induced neurons express *Lumican* and *Crim1*, recently identified to be expressed highly selectively by bulbo-cervical

and thoraco-lumbar CSN, respectively, and to regulate their segmentally specific axon targeting (*Sahni et al., 2021b*; *Sahni et al., 2021a*; *Itoh et al., 2023*). In striking contrast, *Neurog2*-induced neurons express many cardinal CPN genes at higher levels (e.g. *Epha3* and *Satb2*, which both regulate CPN connectivity) (*Alcamo et al., 2008*; *Nishikimi et al., 2011*; *Figure 5D and E*), further reinforcing that *Neurog2* alone is insufficient for appropriate and resolved differentiation of SOX6+/NG2+ progenitors to cortical output identity.

Together, these results highlight that optimized directed differentiation is achieved by emulating normal developmental steps of sequential subtype specification of neocortical neurons regulated by interactions between broad proneural programs and lineage-specific transcription factors with dynamic temporal expression and cross-regulatory activities.

## Discussion

In the work presented here, we first FACS-purify and characterize a subpopulation of postnatal cortical progenitors that are molecularly related to early developmental cortical projection neuron-specific progenitors. We next identify that developmental transcriptional controls can direct the differentiation of SOX6+/NG2+ cortical progenitors into CSN-like neurons *in vitro*. *Fezf2*, a molecular control over SCPN/CSN development, and transcriptional regulators *Neurog2* and *VP16-Olig2* (together, 'NVOF') are able to activate a dormant neurogenic program and overcome the default postnatal gliogenic differentiation program of these cortical progenitors. This directed differentiation generates neurons with a glutamatergic neuronal identity and specific morphologic, molecular, and electrophysiologic features of cortical output neurons resembling corticospinal/subcerebral projection neurons. Our results reveal that NVOF-directed neurons acquire the key molecular features of mature glutamatergic neurons (e.g. expression of NeuN, vGLUT1, CAMK2A, SYN1, SHANK1, and ionotropic and metabotropic glutamate receptors), a cortical projection neuron-like morphology with a single long NF-M+ primary axon and a MAP2+ apical dendrite-like process, the expression of molecular controls specific for SCPN/CSN (e.g. BCL11B/CTIP2, CRYM, EPHB1, and PCP4), and, importantly, do not express molecular markers of alternate fates (e.g. SATB2, BCL11A/CTIP1, CUX1, GABA, DARPP32, TH, 5HT, ISL1). We identify that these critical specifics of differentiation are not reproduced by commonly employed *Neurog2*-driven differentiation. Together, our work indicates that directed differentiation via combinatorial and complementary action of central developmental transcriptional controls enables previously inaccessible specificity in generating defined neuronal subtypes for cellular regeneration or disease modeling of degenerated or damaged neuronal circuitry.

### In contrast to *Neurog2*-only activation, NVOF-directed neurons acquire multimodal CSN identity

The neurons differentiated by NVOF closely resemble *bona fide* corticospinal neurons. Direct transcriptomic comparison with primary SCPN/CSN reveals that NVOF-directed neurons express a large number of SCPN/CSN-enriched genes (*Figure 4J and K*), with close similarity to SCPN/CSN (*R*=0.87) (*Figure 5A*), and their unipolar somatodendritic-axonal morphology also closely resembles that of purified CSN (*Ozdinler and Macklis, 2006*). In contrast to *Neurog2*-induced neurons, NVOF-directed neurons express multiple genes that typically identify CSN specifically. These include the general indicator *Crymu*, as well as *Lumican* and *Crim1*, expressed highly selectively by bulbo-cervical and thoraco-lumbar CSN, respectively, and regulate their segmentally specific axon targeting (*Sahni et al., 2021b*; *Sahni et al., 2021a*; *Itoh et al., 2023*). Quite importantly, NVOF-directed neurons do not display substantial enrichment of key CPN-specific molecular controls (*Figure 5C–E*), indicating that they do not acquire 'mixed,' 'confused' identity. This is all in stark contrast to *Neurog2*-only induced neurons, which display aberrant multipolar morphology, mixed transcriptomic signatures, and substantial co-expression of what are normally developmentally exclusionary differentiation regulators and CPN+SCPN molecular signatures.

While *Neurog2* is expressed dynamically in cortical progenitors during generation of major neuronal subtypes (*Britz et al., 2006*), *Neurog2* knockout does not show significant perturbations to the expression of molecular hallmarks of these neurons (*Hand and Polleux, 2011*; *Dennis et al., 2017*). *Neurog2* misexpression by electroporation during the production of superficial layers does not induce characteristic molecular features of deep layer neurons, although a subset of axons of the

transfected neurons are re-directed to the ventral telencephalon (**Dennis et al., 2017**). Conversely, genetic deletion of *Neurog2* or shRNA knockdown of *Neurog2* from superficial layer intracortical neurons results in variable defects of midline crossing as well as misrouting of callosal axons toward aberrant cortical and subcortical targets (**Hand and Polleux, 2011**). Together, these data suggest that *Neurog2* has only a limited lineage-instructive role over specification of cortical output neurons.

*Neurog2* is also expressed by progenitors of spinal motor neurons, sensory neurons, and dopaminergic neurons in the mammalian brain, and regulates their specification and differentiation (**Hulme et al., 2022**). Therefore, it is conceivable that *Neurog2* expression will induce a subset of its genomic targets depending on the starting cell population, culture conditions, or *in vivo* context. In agreement with this hypothesis, recent reports have identified mixed subtype features in neurons generated from ES/iPS cells by *Neurog2* alone (**Lin et al., 2021**; **Ang et al., 2024**; **Chen et al., 2020**). Several approaches, including pre-patterning of progenitors, combinatorial expression of a cocktail of transcription factors, temporal control of *Neurog2* expression, induction of signaling pathways with small molecules, and co-culture with astrocytes, have been successfully used to sharpen cell fate specification (**Lin et al., 2021**; **Ang et al., 2024**; **Chen et al., 2020**; **Rosa et al., 2020**). Consistent with these results, our NVOF transcriptional regulator combination robustly generates cortical output neuron-like cells compared to *Neurog2* alone.

Intriguingly, even though NVOF-directed neurons acquire both type-specific identity of cortical output neurons, and highly specific indicators of CSN identity, they do not fully resolve the subtype-specific identities of purely subcerebral vs. corticothalamic (CThPN) neurons. They express *Fog2* and *Tbr1*, markers of corticothalamic neurons that are not normally expressed by most mature SCPN/CSN. SCPN and CThPN together comprise cortical output neurons. SCPN and CThPN are located in deep cortical layers V and VI, respectively, and both subtypes send their axons away from cortex via the internal capsule. Not only do these two subtypes share predominant portions of the molecular developmental programs regulating their specification, post-mitotic differentiation, and axon guidance, but approximately 5% are dual SCPN-CThPN that express both high-level *Bcl11b/Ctip2* and *Fog2*, and that send dual projections to both thalamus and subcerebral targets (**Galazo et al., 2016**; **Galazo et al., 2023**). These dual-projecting neurons are thought to 'share' cortical output information with multiple targets for sensorimotor integration. It is possible that the neurons generated here by NVOF-directed differentiation are dual SCPN-CThPN. Recent results identify that the non-DNA-binding transcriptional co-repressor TLE4 forms a complex with transcription factor FEZF2 to epigenetically regulate *Fezf2* expression levels and thus the balance between SCPN and CThPN molecular and projection identity at least through the first postnatal week in mouse (**Galazo et al., 2023**). This delineation between SCPN and CThPN follows multiple earlier regulatory steps, e.g., the control by the transcription factor SOX5 over sequential generation of CThPN and SCPN by progressively de-repressing *Fezf2* expression. Thus, resolution between SCPN and CThPN subtypes normally occurs progressively through late differentiation *in vivo*.

More broadly, differential expression of key controls in terms of both their levels and timing of expression, in addition to combinatorial co-expression with other key regulators, delineates differentiation of cortical projection neurons into progressively distinct subtypes with distinct targets and functional circuitry (**Greig et al., 2013**; **Ozkan et al., 2020**; **Greig et al., 2016**; **Woodworth et al., 2016**; **Galazo et al., 2016**; **McKenna et al., 2011**; **Han et al., 2011**; **Lai et al., 2008**). For example, *Fezf2* and *Ctip2* are expressed more highly by SCPN/CSN relative to CThPN, but both subtypes are severely affected by loss of *Fezf2* function (**Molyneaux et al., 2005**; **Hirata et al., 2004**). In this normal developmental context, the partially unresolved state of NVOF-directed neurons might represent a mid-developmental stage of subtype identity acquisition, since early during normal development many molecular controls are expressed broadly, and their expression progressively resolves over time to produce more highly subtype-restricted expression in postnatal cortex (**Azim et al., 2009b**; **Cederquist et al., 2013**). Consistent with this interpretation, the observed increase of *Ctip2* expression over time by NVOF-directed neurons (**Figure 4I**) suggests ongoing subtype identity refinement.

An additional factor in the incomplete delineation of NVOF-directed neurons into SCPN/CSN might be the constitutive expression of *Neurog2*. *Neurog2* expression is normally dynamically regulated in neural progenitors (**Shimojo et al., 2011**). In addition to its well-established role in activation of proneural genes, *Neurog2* might activate some neuronal subtype-specific genes, such as *Fog2* and *Ctip2* (**Mattar et al., 2008**; **Kovach et al., 2013**). In this context, co-expression of *Fog2* and *Ctip2* by

NVOF-directed neurons might be due, at least in part, to constitutive *Neurog2* expression. To begin to overcome this issue, we applied synthetic modified RNA to enable fine-tuning of both level and temporal dynamics of expression of *Neurog2* and observed robust neuronal induction. The regulation of both level and temporal dynamics of expression during normal development suggests that level- and temporal-controlled expression of *Neurog2* coupled with sustained expression of *Fezf2* (*Fezf2* is expressed constitutively by SCPN/CSN *in vivo*) might enable more refined differentiation of SCPN/CSN from SOX6+/NG2+ progenitors.

Yet another contributing factor to the lack of full SCPN-CThPN delineation of NVOF-directed neurons might be that the basic neuronal induction medium lacks critical extrinsic factors (e.g. diffusible morphogens and growth factors) required for full neuronal maturation and identity refinement. We and others have reported similar but more severe 'stalling' of developmental maturation of ES cell-derived cortical-like neurons under standard culture conditions (*Sadegh and Macklis, 2014*). Supporting this hypothesis, co-culture of NVOF-directed neurons with primary cortical cells (including glia), and in the presence of astrocyte-conditioned medium, improves their survival and both morphological and electrophysiological maturation (*Figure 3*).

Taken together, independent regulation over both level and temporal dynamics of individual transcription factor expression, along with culture in optimized induction medium, might likely generate neurons with even further refined identities and distinction between closely related subtypes.

## SOX6+/NG2+ progenitors are a subset of cortical 'NG2 progenitors' with distinct molecular and functional features

The broad group of cells often collectively characterized by shared expression of NG2 proteoglycan constitute ~2–3% of neural cells in adult rodent cortex, and are the primary proliferative cell group from early postnatal stages through adulthood and in the aged CNS (*Dawson et al., 2003*; *Huang et al., 2020*). Recent work reveals that this broad group of 'NG2 progenitors' is not a homogeneous population; rather, it consists of at least several subpopulations with distinct molecular, cellular, and functional properties (*Viganò and Dimou, 2016*; *Chamling et al., 2021*; *Fang et al., 2023*; *Marisca et al., 2020*; *Spitzer et al., 2019*; *Sánchez-González et al., 2020*; *Janeckova et al., 2024*; *Kirdajova et al., 2021*; *Tsoa et al., 2014*; *Hilscher et al., 2022*; *Floriddia et al., 2020*). While some NG2-expressing progenitors generate oligodendrocytes throughout life, most of them do not differentiate and remain proliferative in the cortex (*Hughes et al., 2013*; *Sánchez-González et al., 2020*). A subset of these cells generates protoplasmic astrocytes in the ventral forebrain and spinal cord (*Zhu et al., 2008a*; *Zhu et al., 2008b*), and a smaller subset has been reported to generate neurons in the piriform cortex (*Rivers et al., 2008*; *Guo et al., 2010*) and dorsolateral cortex (*Janeckova et al., 2024*).

During development, diverse sets of NG2-expressing progenitors arise from anatomically and molecularly distinct dorsal and ventral proliferative zones in sequential waves (*Liu et al., 2021*; *Kessaris et al., 2006*). A substantial proportion of the NG2-expressing progenitors in the cortex (~80% in postnatal rodents) share a common lineage with cortical projection neurons in mice (*Tripathi et al., 2011*) and are thus exposed to the same morphogen gradients and epigenetic landscaping. This shared origin and molecular history provides a strong developmental basis for understanding mechanistically why these SOX6+/NG2+ cortical progenitors that originate from the dorsal (pallial) cortical proliferative zone might be especially competent for directed differentiation into cortical projection neurons, and cortical output neurons in particular.

Of particular note with regard to potential regenerative applications, repopulation of degenerated or injured neurons in particular, SOX6+/NG2+ progenitors, like NG2-expressing progenitors more broadly, are widely distributed in cortex in a tiled manner. Furthermore, progenitors lost due to differentiation or cell death are replenished by cell division and migration of neighboring progenitors (*Hughes et al., 2013*; *Trotter et al., 2010*). Thus, SOX6+/NG2+ progenitors are already positioned local to sites of existing neuron degeneration or other pathology, thus theoretically avoiding the need for long-distance migration and appropriate positioning that would be necessary for transplanted exogenous progenitors, induced neurons, or spatially restricted adult neuronal progenitors from adult neurogenic regions, such as the anterior subventricular zone or hippocampal dentate gyrus. This broad, tiled distribution adds substantially to their potential for cellular repopulation and regenerative approaches.

## Cortical SOX6+/NG2+ progenitors are developmentally poised to generate projection neurons

Our finding that loss of *Sox6* de-represses the proneural gene *Neurog2* strongly indicates that *Sox6* continues to function importantly in regulation of proneural genes in cortical progenitors postnatally, and that SOX6+/NG2+ progenitors actively suppress neurogenic potential. Our observation of *Neurog2* de-repression in the absence of *Sox6* function suggests that downregulation of *Sox6* might be considered as an additional or an alternate molecular regulator for future directed differentiation experiments. Reinforcing this interpretation, even transient expression of *Neurog2* alone via a single dose of synthetic modified mRNA is sufficient to induce TUJ1+ neurons (*Figure 2—figure supplement 5H*), and, upon NVOF expression, substantial numbers of progenitors lose progenitor features and acquire unipolar neuronal morphology by 3 days post-transfection (*Figure 2E*). Furthermore, and quite remarkably, over-expression of the SCPN/CSN-molecular control *Ctip2* (which has no known proneural function) in SOX6+/NG2+ progenitors is sufficient to induce unipolar neuronal morphology, TUJ1 expression, and down-regulation of glial genes (*Figure 2—figure supplement 1F and G*). Together, these results indicate that SOX6+/NG2+ progenitors have substantial competence to differentiate into neurons and that they are at a relatively advanced stage of progenitor fate acquisition.

Directed differentiation of type- or subtype-specific neurons from a developmentally related population of local progenitors might encounter fewer epigenetic blocks than with stem cell or less closely related progenitor populations, thus resulting in improved functional differentiation of type- or subtype-specific neurons (*Herrero-Navarro et al., 2021*). Recent studies have documented that residual transcriptional, epigenetic, and chromatin domain signatures specific to cells of origin persist during derivation of iPSCs, e.g., especially during early passages (*Polo et al., 2010*; *Beagan et al., 2016*; *Krijger et al., 2016*). Such bias and/or blockade is likely to be suboptimal for differentiation of functional type- or subtype-specific neurons, and thus for either functional regeneration or reliable modeling of pathology. Circumstantially supporting this view of persistent effects of cellular origin, reprogramming of fibroblasts to neuronal lineage occurs at a much lower efficiency and more slowly compared to reprogramming of cultured postnatal astrocytes (*Ninkovic and Götz, 2015*), or to our results reported here. Intriguingly, we find that cortical SOX6+/NG2+ progenitors transfected with the single factor *Fezf2* acquire a hybrid morphology, preserving glia-like cell body morphology while developing a neuron-like, single, long primary neurite (*Figure 2—figure supplement 1D and E*). These results suggest incomplete and heterogeneous neuronal induction. Since *Fezf2* has no known proneuronal function, and since it functions centrally in specification and differentiation of cortical output neurons with long axons, it is possible that some of the *Fezf2*'s target genes and their regulatory domains remain epigenetically accessible in cortical SOX6+/NG2+ progenitors. This partial, seemingly hybrid, differentiation driven by *Fezf2* alone further reinforces both the competency of SOX6+/NG2+ progenitors to differentiate relatively efficiently into cortical output projection neurons and the need for multi-component regulation to guide cortical output projection neuron differentiation while suppressing alternative fates and enhancing cell type distinction.

## Summary

The work reported here substantially and uniquely advances the goal of induction of neurogenesis and directed differentiation of subtype-specific neurons from endogenous adult progenitors. We first identify the SOX6+/NG2+ cortical progenitor population and employ genetic access to pure cultures of these progenitors. We then develop NVOF, a first-generation multi-component transcriptional regulatory construct, that induces cortical output neuron-directed differentiation while suppressing the otherwise default glial differentiation pathway. We next identify that NVOF-directed neurons derived from SOX6+/NG2+ cortical progenitors differentiate with remarkable fidelity to *bona fide in vivo* cortical output neurons with appropriate morphological, molecular, deep transcriptomic, and electrophysiological characteristics. Furthermore, these neurons do not display characteristics of alternative neuron types, most notably not even of closely related non-output-neuron cortical projection neurons. This sharp subtype delineation is in striking contrast to previously developed approaches (e.g. fibroblast or iPSC-derived iNs, or glial-derived neuron-like cells) that generate much more 'generic' neuron-like cells with mixed molecular identity, multipolarity, and often continued expression of some genes residual from the cells of origin, further confusing the output cellular identity (*Autar et al., 2022*; *Cao et al., 2017*; *Miskinyte et al., 2017*; *Kim et al., 2010*). Instead, SOX6+/

NG2+ cortical progenitor-derived neurons closely resemble corticospinal/subcerebral projection neurons with some hybrid corticothalamic molecular markers (the two dominant and developmentally closely related subtypes of the specialized cortical output neurons), reminiscent of the ~5% population of CSN/SCPN *in vivo* with hybrid corticothalamic molecular and projection features. Together, this developmentally based directed differentiation from developmentally appropriate adult cortical progenitors sets a precedent and foundation for future optimizations of combinatorial levels, order, temporal dynamics, and subcellular localizations of an appropriate set of molecular controls over subtype-specific neuronal differentiation for *in vitro* mechanistic and therapeutic disease modeling, and toward regenerative neuronal repopulation and circuit repair.

# Materials and methods

## Key resources table

| Reagent type (species) or resource | Designation | Source or reference | Identifiers | Additional information |
|---|---|---|---|---|
| Strain, strain background (*Mus musculus*) | CD1 wild-type | Charles River Laboratories (Wilmington, MA) | | Used for all baseline and crossbreeding experiments |
| Genetic reagent (*M. musculus*) | NG2.DsRed.BAC | Jackson Laboratory | Stock# 008241; RRID:IMSR_JAX:008241 | Generated by *Zhu et al., 2008a*; *Zhu et al., 2008b*; used for isolation of NG2+ progenitors |
| Genetic reagent (*M. musculus*) | Sox6 knockout | Gift from V. Lefebvre (Cleveland Clinic) | | Maintained on C57BL/6 and outcrossed into CD1 background |
| Cell line (*M. musculus*) | NG2-DsRed+ progenitors | This study | | FACS purification from NG2.DsRed transgenic mice. |
| Chemical compound, drug | BrdU | Sigma-Aldrich | B5002 | (50 μg/mg/injection or 1.5 mg/mL in drinking water) |
| other | pCBIG plasmid (*Mus musculus*) | Gift from C. Lois (Caltech) | | CMV/β-actin promoter-driven plasmid for expression constructs |
| Recombinant DNA reagent | NVOF construct (GFP-Neurog2-VP16:Olig2-Fezf2-HA) | This paper | | Created by cloning coding sequences into pCBIG vector with 2A linkers |
| Recombinant DNA reagent | pORFin / pORFinB | D. Rossi Lab (Boston Children's Hospital) | | Vectors used for synthetic mRNA cloning and *in vitro* transcription |
| Chemical compound, drug | Kynurenic acid | Sigma-Aldrich | K3375 | Used in dissection and dissociation medium |
| Chemical compound, drug | DL-2-amino-5-phosphonopentanoic acid (APV) | Sigma-Aldrich | A5282 | Used in dissection and dissociation medium |
| Chemical compound, drug | DL-Cysteine hydrochloride | Sigma-Aldrich | C9768 | Used in enzymatic dissociation medium |
| Chemical compound, drug | Papain | Worthington | LS003126 | Used in cortical dissociation |
| Chemical compound, drug | DNAse I | Sigma-Aldrich | D5025 | Used in enzymatic digestion for dissociation |
| Chemical compound, drug | Poly-D-lysine | Sigma-Aldrich | P0899 | Substrate coating for cell culture |
| Chemical compound, drug | Laminin | Thermo Fisher | 23017015 | Used in cell culture substrate coating |
| Chemical compound, drug | Poly-L-ornithine | Millipore | A-004-C | Used for coating cover glasses |
| Chemical compound, drug | Fugene 6 | Promega | | Transfection reagent for DNA and RNA |
| Chemical compound, drug | PDGF-A | Peprotech | 315–17 | 10 ng/mL in growth medium |
| Chemical compound, drug | FGF2 | Peprotech | 450–33 | 10–20 ng/mL in growth medium |
| Chemical compound, drug | EGF | Peprotech | 315–09 | 20 ng/mL in growth medium |
| Chemical compound, drug | DAPI stain | SouthernBiotech | 0100–20 | Used for nuclear staining |
| Chemical compound, drug | Alexa Fluor 555-conjugated Cholera Toxin | Invitrogen | C22843 | Retrograde labeling of neurons |
| Commercial assay or kit | Superscript IV First-Strand Synthesis System | Thermo Fisher Scientific | 18090050 | Used for cDNA synthesis |
| Chemical compound, drug | Random Hexamers | Thermo Fisher Scientific | SO142 | Used for cDNA priming |
| Commercial assay or kit | iTaq Universal Sybr Green Supermix | Bio-Rad | | Used for qPCR |

*Continued on next page*

*Continued*

| Reagent type (species) or resource | Designation | Source or reference | Identifiers | Additional information |
|---|---|---|---|---|
| Commercial assay or kit | RNeasy Plus Mini Kit | Qiagen | 74134 | RNA isolation with gDNA elimination step |
| Commercial assay or kit | Kapa mRNA HyperPrep Kit | Roche (formerly Kapa Biosystems) | | Used for library prep (14 cycles, PolyA enrichment) |
| Commercial assay or kit | Kapa qPCR Library Quantification Kit | Kapa Biosystems | | Library quantification prior to sequencing |
| Antibody | Anti-ANK3 (ANKYRIN-G) (Mouse monoclonal) | Santa Cruz Biotechnology | sc-12719; RRID:AB_626674 | (1:250) |
| Antibody | Anti-BrdU (Rat monoclonal) | AbD Serotec | OBT0030; RRID:AB_2313756 | (1:500) |
| Antibody | Anti-CSPG4 (NG2) (Rabbit polyclonal) | Millipore | AB5320; RRID:AB_91789 | (1:500) |
| Antibody | Anti-CTIP2 (Rabbit polyclonal) | Abcam | ab28448; RRID:AB_1140055 | (1:500) |
| Antibody | Anti-CTIP2 (Rat monoclonal) | Abcam | ab18465; RRID:AB_2064130 | (1:250) |
| Antibody | Anti-CUX1 (Rabbit polyclonal) | Santa Cruz Biotechnology | sc-13024; RRID:AB_2261231 | (1:200) |
| Antibody | Anti-DARPP32 (Rabbit polyclonal) | Cell Signaling Technology | 2306 S; RRID:AB_823479 | (1:250) |
| Antibody | Anti-FOG2 (Rabbit polyclonal) | Santa Cruz Biotechnology | sc-10755; RRID:AB_2218978 | (1:250) |
| Antibody | Anti-FOXP2 (Rabbit polyclonal) | Abcam | ab16064; RRID:AB_2314424 | (1:2000) |
| Antibody | Anti-GABA (Mouse monoclonal) | Sigma-Aldrich | A0310; RRID:AB_476667 | (1:200) |
| Antibody | Anti-GFAP (Mouse monoclonal) | Sigma-Aldrich | G3893; RRID:AB_477010 | (1:1000) |
| Antibody | Anti-GFAP (Rabbit polyclonal) | Sigma-Aldrich | G9269; RRID:AB_477035 | (1:1000) |
| Antibody | Anti-GFP (Chicken polyclonal) | Invitrogen | A10262; RRID:AB_2534023 | (1:1000) |
| Antibody | Anti-GFP (Rabbit polyclonal) | Invitrogen | A11122; RRID:AB_221569 | (1:1000) |
| Antibody | Anti-HA (Mouse monoclonal) | Covance | MMS-101R; RRID:AB_291262 | (1:1000) |
| Antibody | Anti-ISL1 (Mouse monoclonal) | Novus | H00003670; RRID:AB_539948 | (1:250) |
| Antibody | Anti-MAP2 (Mouse monoclonal) | Sigma | M1406; RRID:AB_477171 | (1:500) |
| Antibody | Anti-NESTIN (Chicken polyclonal) | Novus | NB100-1604; RRID:AB_2282642 | (1:2000) |
| Antibody | Anti-NeuN (Mouse monoclonal) | Chemicon | MAB377; RRID:AB_2298772 | (1:500) |
| Antibody | Anti-NF-M (Rabbit polyclonal) | Millipore | AB1987; RRID:AB_91201 | (1:200) |
| Antibody | Anti-NEUROG2 (Mouse monoclonal) | R&D Systems | MAB3314; RRID:AB_2149520 | (1:100) |
| Antibody | Anti-OLIG2 (Goat polyclonal) | R&D Systems | AF2418; RRID:AB_2157554 | (1:200) |
| Antibody | Anti-RFP (Rat monoclonal) | Antibodies-online | ABIN334653; RRID:AB_10795839 | (1:500) |
| Antibody | Anti-PCP4 (Rabbit polyclonal) | Proteintech | 14705–1-AP; RRID:AB_2878075 | (1:500) |
| Antibody | Anti-PDGFRB (Rabbit polyclonal) | Cell Signaling | 3169; RRID:AB_2878075 | (1:100) |
| Antibody | Anti-PSA-NCAM (Mouse monoclonal) | Chemicon | MAB5324; RRID:AB_95211 | (1:200) |
| Antibody | Anti-SATB2 (Mouse monoclonal) | Abcam | ab51502; RRID:AB_882455 | (1:200) |
| Antibody | Anti-SATB2 (Rabbit polyclonal) | Abcam | ab34735; RRID:AB_2301417 | (1:500) |
| Antibody | Anti-SOX6 (Rabbit polyclonal) | Abcam | ab30455; RRID:AB_1143033 | (1:500) |
| Antibody | Anti-SOX10 (Goat polyclonal) | Santa Cruz | sc-17342; RRID:AB_2195374 | (1:200) |
| Antibody | Anti-SYNAPSIN (Rabbit polyclonal) | Synaptic Systems | 106 002; RRID:AB_887804 | (1:500) |
| Antibody | Anti-SYNAPTOPHYSIN (Mouse monoclonal) | Millipore | MAB5258; RRID:AB_2313839 | (1:500) |
| Antibody | Anti-TH (Rabbit polyclonal) | Millipore | AB152; RRID:AB_390204 | (1:250) |
| Antibody | Anti-TUBB3 (Tuj1) (Rabbit polyclonal) | Sigma | T2200; RRID:AB_262133 | (1:1000) |
| Antibody | Anti-TUBB3 (Tuj1) (Mouse monoclonal) | Biolegend | MMS-435P; RRID:AB_2313773 | (1:1000) |
| Antibody | Anti-vGLUT1 (Rabbit polyclonal) | Synaptic Systems | 135 302; RRID:AB_887877 | (1:500) |

*Continued on next page*

*Continued*

| Reagent type (species) or resource | Designation | Source or reference | Identifiers | Additional information |
|---|---|---|---|---|
| Antibody | Anti-2A peptide (Rabbit polyclonal) | Millipore | ABS31; RRID:AB_11214282 | (1:1000) |
| Antibody | Anti-5HT (Rabbit polyclonal) | Immunostar | 20080; RRID:AB_572263 | (1:3000) |
| Antibody | Alexa-Fluor-conjugated Secondary Antibodies (various hosts) | Invitrogen | | (1:1000); Used for ICC |
| Other | Positive Control tissue samples | This paper | | Used to validate ICC |
| Software, algorithm | Nikon NIS Elements | Nikon | RRID:SCR_014329 | Image acquisition and quantification |
| Software, algorithm | GraphPad Prism 8 | GraphPad | RRID:SCR_002798 | Statistical analysis |
| Software, algorithm | RStudio (v1.3.959) | RStudio | RRID:SCR_000432 | Data analysis and visualization |
| Software, algorithm | FASTQC | Babraham Institute | RRID:SCR_014583 | Sequencing quality control |
| Software, algorithm | STAR | *Dobin et al., 2013* | RRID:SCR_015899 | Alignment of RNA-seq reads |
| Software, algorithm | DESeq2 | *Love et al., 2014* | RRID:SCR_015687 | Differential expression analysis |
| Software, algorithm | PANTHER database | *Mi et al., 2021* | RRID:SCR_004869 | GO enrichment analysis |
| Other | Nanoject II | Drummond | | Retrograde labeling of neurons |
| Other | Vevo 770 ultrasound backscatter microscopy system | VisualSonics | | Retrograde labeling of neurons |
| Other | FACSAria II Cell sorter | Becton Dickinson | | Isolation of NG2-DsRed+progenitors |
| Other | Aspirator Tube for electroporation | Sigma | A5177 | Plasmid *in utero* electroporation |
| Other | CUY21edit Electroporator | Bex Co. Ltd | | Plasmid *in utero* electroporations |

## Mice

All mouse studies were approved by the Harvard University IACUC and were performed in accordance with institutional and federal guidelines. The date of vaginal plug detection was designated embryonic day (E) 0.5, and the day of birth as postnatal day (P) 0. Wild-type CD1 mice were purchased from Charles River Laboratories (Wilmington, MA). The NG2.DsRed.BAC mouse line was generated by Nishiyama and colleagues (*Zhu et al., 2008a*) and was procured from Jackson Laboratories (stock number: 008241). *Sox6* knockout mouse was the generous gift of V. Lefebvre (Cleveland Clinic) (*Smits et al., 2001*) and was maintained on a C57BL/6 background and separately crossed into an outbred CD1 background. Most *Sox6* knockout embryos on a c57 background die perinatally (*Azim et al., 2009a*), while outcrossing into the CD1 background resulted in live *Sox6* knockout pups. These pups survived for several days, developed poor body condition, and died by about P14. Male and female pups were included in all retrolabeling, FACS purification, and culture experiments. All mice were maintained in standard housing conditions on a 12 hr light/dark cycle with food and water *ad libitum*. A maximum of four adult animals were housed per cage.

All mouse studies were approved by the Harvard University IACUC (protocol numbers HU IACUC # 11-19-4 and HU IACUC ID # 11-22-2) and were performed in accordance with institutional and federal guidelines.

## BrdU labeling

To cumulatively label dividing cells in the cortex at P7 and P28, BrdU (Sigma, B5002) was injected intraperitoneally from P3 to P7 or from P23 to P28 (50 µg/mg/injection). To cumulatively label slowly dividing and/or quiescent populations in adult brain, BrdU was added to drinking water for 4–6 weeks (1.5 mg/mL). Brains were collected at corresponding ages and processed for BrdU immunocytochemistry.

## Plasmids

CMV/β-actin promoter-driven plasmid pCBIG (derived from CBIG, a gift from C. Lois, Caltech) was used to drive expression of IRES-GFP (control), single factors (*Ctip2*, *Neurog2*, *VP16:Olig2*, *Fezf2-HA*, and tdTomato) or NVOF construct. The NVOF construct was created by cloning GFP, *Neurog2*, *VP16-Olig2*, and *Fezf2-HA* coding sequences separated by 2A linker sequences into a pCBIG vector (*Supplementary file 3*). In this system, genes linked to each other via viral 2A sites are transcribed as

a single mRNA, but are translated into individual polypeptides (*Tang et al., 2009*; *Donnelly et al., 2001*; *Szymczak et al., 2004*). For synthetic modified mRNA synthesis, GFP, RFP, *Fezf2-HA*, and *Neurog2* open reading frames were cloned into pORFin or pORFinB vectors (from D. Rossi Lab, HSCRB and Boston Children's Hospital). pORFin vectors had the appropriate 5' and 3' UTR sequences flanking the cloning sites, and an upstream T7 promoter for *in vitro* transcription. RNA was synthesized in accordance with a published protocol (*Mandal and Rossi, 2013*).

## Purification and culture of cortical SOX6+/NG2+ progenitors

Heterozygous offspring pups (P2-P5) from the NG2-DsRed male and wild-type CD1 female crosses were used for FACS experiments. Pups were screened for red fluorescence under a dissecting microscope (Nikon, SMZ-1500) and anesthetized on ice. Brains were dissected, and meninges were removed in ice-cold Hank's buffered salt solution (HBSS) (Gibco, 14025092). Neocortices were micro-dissected in ice-cold dissociation medium (pH 7.35), composed of 20 mM glucose (Sigma, G6152), 0.8 mM kynurenic acid (Sigma, K3375), 0.05 mM DL-2-amino-5-phosphonopentanoic acid (APV) (Sigma, A5282), 100 U/ml penicillin - 100 μg/ml streptomycin (Gibco, 15140122), 0.09 M $Na_2SO_4$, 0.03 M $K_2SO_4$, and 0.014 M $MgCl_2$ (pH 7.35±0.02). Dissected cortices were enzymatically digested in dissociation medium containing 0.16 mg/ml DL-Cysteine hydrochloride (Sigma, C9768), 10 U/ml papain (Worthington, LS003126), and 30 U/ml DNAse I (Sigma, D5025) at 37 °C for 30 min, rinsed two times with ice-cold OptiMEM (Gibco, 51985034), and supplemented with 20 mM glucose, 0.4 mM kynurenic acid, and 0.025 mM APV to protect against glutamate-induced neurotoxicity (*Catapano et al., 2001*). Digested cortices were mechanically dissociated by gentle trituration using fire-polished glass Pasteur pipets to create a single-cell suspension. Dissociated cells were centrifuged at 100 g for 5 min at 4 °C, resuspended (5–10×10⁶ cell/ml) in OptiMEM with supplements, and filtered through a 35 μm cell strainer (Corning, 352235). All chemicals were purchased from Sigma-Aldrich unless stated otherwise.

Cells were purified based on DsRed fluorescence intensity using a BD FACSAria II cell sorter in four-way purity mode (85 μm nozzle). DsRed-positive cells from the NG2.DsRed BAC-transgenic mouse cortex consisted of two distinct populations: bright and dim. After qPCR and immunocytochemical characterization of both populations, only the bright population, which yielded 200–300K cells/brain, was purified for induced neurogenesis experiments. A previously published protocol was adapted to maintain cells in a proliferative progenitor state (*Najm et al., 2013*). Purified cells were sorted into and cultured in growth medium, composed of DMEM/F12 with GlutaMAX (Gibco, 10565018), 15 mM HEPES (Gibco, 15630106), B27 without vitamin A (Gibco, 12587–010), N2-max (R&D Systems, AR009), 100 U/ml penicillin - 100 μg/ml streptomycin (Gibco, 15140122), 10 ng/ml PDGF-A (Peprotech, 315–17), and 20 ng/ml FGF2 (Peprotech, 450–33). Half of the medium in each well was replaced every other day. Cells were seeded (~10 K cells/cm²) on either 50–100 μg/ml poly-D-lysine (Sigma, P0899) plus laminin (Thermo, 23017015), or 0.01% poly-L-ornithine (Millipore, A-004-C) plus laminin-coated cover glasses (Fisher, 12-545-81) in 24-well plates for microscopy experiments (Corning, 353047), or without cover glass in 6-well plates for RNA experiments (Corning, 353047). Transfection was performed at ~5 DIV after half-replacing the medium with fresh proliferation medium using Fugene 6 (Promega) with the following ratio: per 6-well plate, 600 μl DMEM/F12 medium (w/o supplements), 30 μl transfection reagent, and 8 μg of DNA was mixed, incubated for 15–30 min, and directly added into each well (~100 μl/well), yielding ~10% transfection rate at 24 hr. The same Fugene 6 transfection reagent was used for synthetic RNA transfections (20 μl media, 1.2 μl transfection reagent, and 0.2 μg RNA for each well of the 24-well plate). On the day following transfection, growth medium was replaced with neuronal induction medium, composed of a 1:1 mixture of DMEM/F12 and Neurobasal-A (Gibco, 10888022), GlutaMAX (Gibco, 35050061), 15 mM HEPES, B27 with vitamin A (Gibco, 17504044), N2 (Gibco, 17502048), and 100 U/ml penicillin - 100 μg/ml streptomycin (Gibco, 15140122). Medium was half-replaced every third day after transfection until fixation.

## Retrograde labeling and FACS purification of SCPN and CPN

Retrograde labeling experiments were adapted from previously published procedures (*Arlotta et al., 2005*). Briefly, pups were anesthetized by hypothermia at P0/P1, and SCPN and CPN were retrolabeled from their corresponding axonal projections by pressure injection (Nanoject II, Drummond) of Alexa Fluor 555-conjugated cholera toxin, subunit B (CTB) (Invitrogen, C22843) (6–7 injections, 23 nl/injection, 2 μg/ul) using pulled and beveled glass micropipettes with a tip diameter of 30–50 μm. SCPN

were labeled from the cerebral peduncle, and CPN were labeled from contralateral corpus callosum close to the midline (3–4 rostrocaudal levels). Injections were performed in deeply anesthetized pups using a Vevo 770 ultrasound backscatter microscopy system (VisualSonics). Brains were collected at P2 for FACS purification, and retrograde labeling success was verified under a fluorescence-equipped dissecting microscope (SMZ-1500; Nikon). Cells were purified with stringent fluorescence gating using a BD FACSAria II cell sorter (85 µm nozzle) in four-way purity mode.

### *In utero* electroporation

Timed pregnant CD1 dams were anesthetized with isoflurane, and an incision was made in the abdomen. The uterine horns were exposed and gently positioned on a sterile piece of gauze. 1.0 µg/µl of plasmid DNA was mixed with 0.005% Fast Green in sterile PBS and injected *in utero* into one lateral ventricle of each embryonic brain. The injections were performed with beveled glass micropipettes (tip diameter of 30–60 µm) via mouth pipetting with an aspirator tube assembly (Sigma, A5177). Plasmid electroporations were performed by placing a positive electrode (tweezer electrodes, 5 mm diameter) above the cortex and a negative electrode behind the head, and applying five pulses of current at 40 V for 50 milliseconds per pulse with 1 s intervals between pulses (CUY21Edit Electroporator, Bex Co. Ltd.). Brains were collected at P7 for NVOF misexpression analysis and at P0-P1 for primary neuron culture.

### Astrocyte-conditioned media

Production of astrocyte-conditioned media was based on the published protocol for primary culture of postnatal cortical astrocytes (*Heinrich et al., 2011*). Briefly, cerebral cortices were micro-dissected from wild-type P5-P7 CD1 pups, gently dissociated without enzymatic digestion using fire-polished glass Pasteur pipets, and centrifuged at 100 g for 5 min at 4 °C. Dissociated cells were seeded in T25 flasks and cultured in astrocyte growth medium DMEM/F12 with GlutaMAX (Gibco, 10565018), 10% fetal calf serum (Seradigm, 97068–091), 5% horse serum (Invitrogen, 26050070), B27 (with vitamin A), 100 U/ml penicillin – 100 µg/ml streptomycin (Gibco, 15140122), 10 ng/ml EGF (Peprotech, 315–09), and 10 ng/ml FGF2 (Peprotech, 450–33). Medium was fully changed 24 hr post-culturing, and half of the medium was replaced three days post-culturing. Culture fidelity was verified by morphology and GFAP expression of the differentiated cells. To obtain astrocyte-conditioned media, astrocytes were passaged at ~5 DIV using trypsin (Gibco, 25200056), centrifuged at 100 g for 5 min at room temperature, diluted 1:4, re-seeded in T75 flasks containing astrocyte growth medium, and cultured for 24 hr. Growth medium was subsequently replaced with neuronal induction medium (described above). The conditioned medium was collected at days 10, and 20, and aliquots were stored at –80 °C.

### NVOF-induced and primary neuron co-culture

To co-culture induced neurons with primary neurons, primary forebrain neurons were obtained from P0-P1 CD1 wild-type pups using the dissociation protocol described above, and directly added onto progenitor cell cultures at 24 hr after transfection (25 K/cm$^2$). One-half of the medium was replaced with fresh astrocyte-conditioned media every third day. For dendritic morphology comparison, cortical projection neurons were labeled via *in utero* electroporation (at E14.5) of a tdTomato reporter plasmid driven by CMV-beta-actin promoter. Neurons were dissociated at P0-P1, cultured in 24-well plates with cover glass (50 K cell/cm$^2$), and cultured in parallel with induced neurons using the same neuronal media that is described above.

### Histology and immunocytochemistry

Immunocytochemistry (ICC) for tissue sections was performed following standard protocols. Briefly, mice were transcardially perfused with PBS then 4% PFA, dissected, and post-fixed overnight at 4 °C in 4% paraformaldehyde. Brains were embedded in 4% low melting temperature agar (Sigma-Aldrich) and sectioned at 50 µm on a vibrating microtome (Leica). Fixed tissues were stored in PBS with 0.025% sodium azide. Floating sections were blocked with 0.3% BSA (wt/vol) (Sigma, A3059), 0.3% Triton X-100 (Sigma, T8787), and 0.025% sodium azide (Sigma, S2002) in PBS for 30 min. Primary antibodies were diluted in the same blocking solution and incubated with sections for 4 hr at room temperature, or overnight at 4 °C. Sections were rinsed three times with PBS for 10 min and incubated with appropriate secondary antibodies diluted in blocking solution for 2–3 hr at room temperature. Sections

were rinsed three times with PBS, and mounted using Fluoromount with DAPI (SouthernBiotech, 0100–20) for image acquisition. ICC for BrdU was preceded by 2 hr of treatment with 2 N HCl at room temperature for antigen retrieval.

ICC for cultured cells was performed by first fixing cells in 4% paraformaldehyde at room temperature for 10 min, rinsing three times with PBS, and storing in PBS with 0.025% sodium azide at 4 °C. Cells were blocked in the blocking solution for 15 min, incubated with primary antibodies for 2 hr, rinsed with PBS three times for 5 min, incubated with secondary antibodies for 45 min, rinsed with PBS three times for 5 min (all reactions at room temperature), and mounted using Fluoromount with DAPI.

The following primary antibodies and dilutions were used: mouse anti-ANK3 (ANKYRIN-G), 1:250 (Santa Cruz, sc-12719); rat anti-BrdU, 1:500 (AbD Serotec, OBT0030); rabbit anti-CSPG4 (NG2), 1:500 (Millipore, AB5320); rabbit anti-CTIP2, 1:500 (Abcam, ab28448); rat anti-CTIP2, 1:250 (Abcam, ab18465); rabbit anti-CUX1, 1:200 (Santa Cruz Biotechnology, sc-13024); rabbit anti-DARPP32, 1:250 (Cell Signaling Technology, 2306 S); rabbit anti-FOG2, 1:250 (Santa Cruz Biotechnology, sc-10755); rabbit anti-FOXP2, 1:2000 (Abcam, AB16064); mouse anti-GABA, 1:200 (Sigma, A0310); mouse anti-GFAP, 1:1000 (Sigma, G3893); rabbit anti-GFAP, 1:1000 (Sigma, G9269); chicken anti-GFP, 1:1000 (Invitrogen, A10262); rabbit anti-GFP, 1:1000 (Invitrogen, A11122); mouse anti-HA, 1:1000 (Covance, MMS-101R); mouse anti-ISL1, 1:250 (Novus, H00003670); mouse anti-MAP2, 1:500 (Sigma, M1406); chicken anti-NESTIN, 1:2000 (Novus, NB100-1604); mouse anti-NeuN, 1:500 (Chemicon, MAB377); rabbit anti-NF-M, 1:200 (Millipore, AB1987); mouse anti-NEUROG2, 1:100 (R&D Systems; MAB3314); goat anti-OLIG2, 1:200 (R&D Systems, AF2418); rat anti-RFP, 1:500 (antibodies-online, ABIN334653); rabbit anti-PCP4, 1:500 (Proteintech, 14705–1-AP); rabbit anti-PDGFRB, 1:100 (Cell Signaling, 3169); mouse anti-PSA-NCAM, 1:200 (Chemicon, MAB5324); mouse anti-SATB2, 1:200 (Abcam, ab51502); rabbit anti-SATB2, 1:500 (Abcam, ab34735); rabbit anti-SOX6, 1:500 (Abcam, AB30455); goat anti-SOX10, 1:200 (Santa Cruz, sc-17342); rabbit anti-SYNAPSIN, 1:500 (Synaptic Systems, 106002); mouse anti-SYNAPTOPHYSIN, 1:500 (Millipore, MAB5258); rabbit anti-TH, 1:250 (Millipore, AB152); rabbit anti-TUBB3 (TUJ1), 1:1000 (Sigma, T2200); mouse anti-TUBB3 (TUJ1), 1:1000 (Biolegend, MMS-435P), rabbit anti-vGLUT1, 1:500 (Synaptic Systems, 135302); rabbit anti-2A-peptide, 1:1000 (Millipore, ABS31), rabbit anti-5HT, 1:3000 (Immunostar, 20080). Alexa Fluor-conjugated secondary antibodies (Invitrogen) were used at a dilution of 1:1000. Positive controls were included in all ICC experiments with negative results. All ICC experiments utilized different batches of FACS-purified cells from independent litters to yield a minimum of three true biological replicates. Primary data were analyzed by one investigator (AO), then confirmed by a second independent investigator (HP).

## Image acquisition, quantification, and statistical analysis

Wide-field image acquisition was performed with a Nikon 90i epifluorescence microscope equipped with a Clara DR-328G cooled CCD digital camera (Andor Technology) running NIS Elements software (Nikon). Brightfield images were acquired using a Nikon ECLIPSE Ts2R-FL inverted microscope. For optimal data visualization, images were adjusted for contrast, brightness, and size in Adobe Photoshop and Illustrator (2019). Identical procedures were applied across different experimental conditions. For cell quantifications, a cover glass area of ~50 $mm^2$ (7×7 tile) was imaged using a 10x objective. The acquired image was binned as 1 $mm^2$ boxes, individual boxes were randomly selected, and all GFP+ cells in each selected box were quantified using NIS-elements software (Nikon). To quantify the immunofluorescence intensity of target molecules, nuclei were identified via DAPI, and the average intensity of the outlined nuclear area was measured on Nikon-NIS. The following criteria were used to mark neurons with multiple axons: If the second longest neurite originating from the cell soma was at least half the length of the longest neurite, that cell was marked as multipolar. A minimum of four independent biological replicates were used for each experimental condition across the study unless otherwise mentioned in the text. Microsoft Excel, RStudio (version 1.3.959), and GraphPad Prism 8 were used for data analysis, plotting graphs, and statistics. Statistical details of the experiments can be found in the figure legends. Significance is based on the p value indicated on the graphs as * p % 0.05, ** p % 0.01, ***p % 0.001, ****p % 0.0001.

## Electrophysiology

Electrophysiological recordings were performed at 20–25°C on an Olympus BX51WI microscope. Cells were bathed in artificial cerebral spinal fluid (ACSF) containing 119 mM NaCl, 2.5 mM KCl, 4 mM

$CaCl_2$, 4 mM $MgSO_4$, 1 mM $NaH_2PO_4$, 26.2 mM $NaHCO_3$, and 11 mM glucose. ACSF was continuously saturated with 95% $O_2$/5% $CO_2$. Intracellular recordings were obtained using glass micropipettes filled with an internal solution containing 136 mM $KMeSO_3$, 17.8 mM HEPES, 0.6 mM $MgCl_2$, 1 mM EGTA, 4 mM Mg-ATP, and 0.3 mM Na-GTP. Traces were collected using a Multiclamp 700B amplifier (Molecular Devices), filtered with a 2 kHz Bessel filter, digitized at 50 kHz using a Digidata 1440 A digitizer (Molecular Devices), stored using Clampex 10 (Molecular Devices), and analyzed off-line via customized procedures written in Igor Pro (WaveMetrics). Series resistance was monitored throughout the experiment. Cells at DPI/DIV 15–16 were identified visually by fluorescence. Action potentials were evoked by injection of current steps, ranging from –140 pA to 400 pA in 60 pA increments, with a duration of 600 ms. Action potential parameters were quantified for the first action potential evoked at the lowest current injection that resulted in an action potential. The threshold potential was defined as the voltage at which dV/dt of the action potential waveform reached 10% of its maximum value, relative to a dV/dt baseline taken 10 ms before the peak. Action potential amplitude was defined as the difference between the threshold value (in mV) and the maximum voltage of the action potential. Width was measured at half-maximum amplitude. Sag current was measured during a –140 pA step current for a duration of 600 ms.

## RNA sequencing

A minimum of three independent biological replicates was used for each experimental group (i.e. mouse litters, cell culture batches, FACS purifications, etc. were different for each biological replicate). RNA isolation was performed using a Qiagen RNeasy Plus Mini Kit with the gDNA elimination step. FACS-purified cells were collected directly into RLT Plus buffer with β-mercaptoethanol. RNA concentration, purity, and integrity were measured by a Nanodrop (Thermo Fisher), an Agilent TapeStation 2200, and an Agilent Bioanalyzer 2100. Only high-quality RNA samples were used for library preparation. For the 32 samples used in this study, the minimum RNA integrity number (RIN) was 8, the average was 9.7, and the median was 10.

Library preparation and sequencing were performed by the Bauer Core Facility at Harvard University. RNA was fragmented at 94 °C for 6 min with a final size range of 200–300 bp. The library was prepared from 50 ng total input RNA per sample using a Kapa mRNA HyperPrep kit (14 cycles) with PolyA enrichment (stranded via dUTP addition, and first-strand preserved). Unique dual 8 bp adapters (1.5 µM) (IDT for Illumina) were used for indexing. The library quality and concentration were confirmed by an Agilent TapeStation 2200 and a Kapa qPCR library quantification kit. The pooled samples were run on Illumina NextSeq High flow cells (75 bp, paired-end reading). Sequencing quality was assessed by FASTQC (version 0.11.9). STAR-aligned counts were used for quality control metrics (*Steinbaugh et al., 2018*).

The quasi-aligned counts from Salmon with default options were used to perform downstream gene expression analyses (*Patro et al., 2017*). Transcript-level count matrices were produced via the Bioconductor package 'tximport' (*Soneson et al., 2015*). Ensembl gene IDs were generated using the GRCm38 reference genome (Ensembl v98). DESeq2 was used to perform differential expression analyses (*Love et al., 2014*). Low count genes (total reads <10) were pre-filtered before DESeq2 functions. Gene names and other information were annotated using the Bioconductor package 'AnnotationDbi.' Variance-stabilizing transformed (vst) normalized counts (log2 scale) were used for data visualization (*Love et al., 2016*). The code used to perform subsequent analyses of the sequencing data was an adaptation of standard R packages. Gene ontology (GO) enrichment analysis was performed using the PANTHER online database (*The Gene Ontology Consortium, 2019*; *Ashburner et al., 2000*). Raw FASTQ files and processed counts are available at https://doi.org/10.7910/DVN/IODOK1.

## Quantitative PCR

cDNA was prepared using the Superscript IV first-strand synthesis system (Thermo, 18090050) and random hexamers (Thermo, SO142) following the manufacturer's standard protocol. Random hexamers were used for amplification. qPCR was performed using the iTaq Universal Sybr Green Supermix (Bio-Rad) on a Bio-Rad CFX96 thermal cycler following standard procedures. For all qPCR primers used in this study, reaction efficiency was calculated by standard curve analysis, and only primers with high efficiency (90–105%) were used. See *Supplementary file 2* for the primer list. Four independent biological replicates were used for each experimental group in all qPCR experiments.

## Materials availability statement

The NVOF construct used in this study can be requested from the laboratory of the corresponding author, Jeffrey D. Macklis (jeffrey_macklis@harvard.edu). The map and sequence of this construct is provided in *Supplementary file 3*.

## Acknowledgements

This work was supported by NINDS grants NS045523, DP1 NS106665, and NS049553, by the Emily and Robert Pearlstein Fund for Nervous System Repair, and by the Max and Anne Wien Professor of Life Sciences fund (to JDM). AO was partially supported by a fellowship from the Suna and Inan Kirac Foundation. HP was partially supported by an International Brain Research Organization Fellowship and a McKnight Brain Research Institute/Regeneration Project Fellowship. We thank Jessica Kim, Jessica Wooten, Ioana Florea, and Ryan Humphries for technical assistance; David Dombkowski at MGH and Girijesh Buruzula, Joyce LaVecchio, and Silvia Ionescu at HSCRB for their help with FACS purification; Andrew Thompson for help with the cloning; Vibhu Sahni and Maria Galazo for assistance with retrograde labeling of SCPN/CSN and scientific discussions; Wataru Ebina for help with synthetic modified RNA experiments; Pratibha Tripathi for advice on astrocyte culture; and other members of the Macklis Laboratory for helpful suggestions and critical reading of the manuscript. This work is dedicated to the memory and intellectual curiosity of Byron Wien.

## Additional information

### Funding

| Funder | Grant reference number | Author |
|---|---|---|
| National Institute of Neurological Disorders and Stroke | NS045523 | Jeffrey D Macklis |
| National Institute of Neurological Disorders and Stroke | DP1 NS106665 | Jeffrey D Macklis |
| National Institute of Neurological Disorders and Stroke | NS049553 | Jeffrey D Macklis |
| Emily and Robert Pearlstein Fund for Nervous System Repair | | Jeffrey D Macklis |
| Max and Anne Wien Professor of Life Sciences Fund | | Jeffrey D Macklis |
| Suna and Inan Kirac Foundation | | Abdulkadir Ozkan |
| International Brain Research Organization | | Hari K Padmanabhan |
| McKnight Brain Research Institute | Regeneration Project Fellowship | Hari K Padmanabhan |

The funders had no role in study design, data collection and interpretation, or the decision to submit the work for publication.

### Author contributions

Abdulkadir Ozkan, Data curation, Formal analysis, Validation, Investigation, Visualization, Methodology, Writing – original draft, Writing – review and editing; Hari K Padmanabhan, Conceptualization, Data curation, Formal analysis, Validation, Investigation, Methodology, Writing – original draft, Writing – review and editing; Seth L Shipman, Investigation, Visualization, Methodology, Writing – original draft; Eiman Azim, Conceptualization, Writing – review and editing; Priyanka Kumar, Data

curation, Investigation, Methodology; Cameron Sadegh, Investigation, Methodology, Writing – review and editing; A Nazli Basak, Resources, Investigation; Jeffrey D Macklis, Conceptualization, Resources, Formal analysis, Supervision, Funding acquisition, Project administration, Writing – review and editing, Writing – original draft

### Author ORCIDs
Abdulkadir Ozkan https://orcid.org/0000-0003-2499-7637
Hari K Padmanabhan https://orcid.org/0000-0002-2289-0249
Eiman Azim https://orcid.org/0000-0002-1015-1772
Jeffrey D Macklis https://orcid.org/0000-0003-3662-9698

### Ethics
All mouse studies were approved by the Harvard University IACUC (protocol numbers HU IACUC # 11-19-4 and HU IACUC ID # 11-22-2) and were performed in accordance with institutional and federal guidelines.

Reviewer #1 (Public review): https://doi.org/10.7554/eLife.100340.3.sa1
Reviewer #2 (Public review): https://doi.org/10.7554/eLife.100340.3.sa2
Author response https://doi.org/10.7554/eLife.100340.3.sa3

---

## Additional files

### Supplementary files
Supplementary file 1. Transcript counts for top-500 genes enriched in major cell lineages (*Zhang et al., 2014*) and mural cells (*He et al., 2016*) for acutely sorted DsRed-Negative, -Dim, and -Bright cells, as well as cultured DsRed-Bright cells (5-DIV). Counts are variance-stabilizing transformed (vst) normalized data in log2 scale.

Supplementary file 2. List of primer sets used in qPCR experiment.

Supplementary file 3. The map and sequence of the NVOF construct.

MDAR checklist

### Data availability
Raw FASTQ files and processed counts are available at https://doi.org/10.7910/DVN/IODOK1.

The following dataset was generated:

| Author(s) | Year | Dataset title | Dataset URL | Database and Identifier |
| --- | --- | --- | --- | --- |
| Ozkan A, Padnabhan H, Macklis JD | 2024 | Directed differentiation of functional corticospinal-like neurons from endogenous SOX6+/NG2+ cortical progenitors | https://doi.org/10.7910/DVN/IODOK1 | Harvard Dataverse, 10.7910/DVN/IODOK1 |

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
