## [Editor Report · eLife Assessment]

This study presents **fundamental** new findings introducing a new approach for the reprogramming of brain glial cells to corticospinal neurons. The data is highly **compelling**, with multiple lines of evidence demonstrating the success of this new assay. These exciting findings set the stage for future studies of the potential of these reprogrammed cells to form functional connections *in vivo* and their utility in clinical conditions where corticospinal neurons are compromised.

---

## [Referee Report · Reviewer #1 (Public review)]

Summary:

The manuscript by Ozcan et al., presents compelling evidence demonstrating the latent potential of glial precursors of the adult cerebral cortex for neuronal reprogramming. The findings substantially advance our understanding of the potential of endogenous cells in the adult brain to be reprogrammed. Moreover, they describe a molecular cocktail that directs reprogramming toward corticospinal neurons (CSN).

Strengths:

Experimentally, the work is compelling and beautifully designed. The work provides a characterization of endogenous progenitors, genetic strategies to isolate them, and proof of concept of exploiting these progenitors' potential to produce a specific desired neuronal type with "a la carte" combination of transcription factors.

Weaknesses:

This study demonstrates reprogramming *in vitro*. Future research will need to assess how these reprogrammed corticospinal neurons integrate and function under physiological conditions and in models of trauma or neurodegeneration.

Although still in its early stages, neural reprogramming holds significant promise. This study reinforces the hope that, in the future, it may be possible to restore lost or damaged neurons through targeted cellular reprogramming.

---

## [Referee Report · Reviewer #2 (Public review)]

Summary:

Here the authors show a novel direct neuronal reprogramming model using a very pure culture system of oligodendrocyte progenitor cells and demonstrate hallmarks of corticospinal neurons to be induced when using Neurogenin2, a dominant-negative form of Olig2 in combination with the CSN master regulator Fezf2.

Strengths:

This is a major achievement as the specification of reprogrammed neurons towards adequate neuronal subtypes is crucial for repair and is still largely missing. The work is carefully done, and the comparison of the neurons induced only by Neurogenin 2 versus the NVOF cocktail is very interesting and convincingly demonstrates a further subtype specification by the cocktail.

Weaknesses:

As carefully as it is done *in vitro*, the identity of projection neurons can best be assessed *in vivo*. If this is not possible, it could be interesting to co-culture different brain regions and see if these neurons reprogrammed with the cocktail, indeed preferentially send out axons to innervate a co-cultured spinal cord versus other brain region tissue.

---

## [Author Response]

The following is the authors’ response to the original reviews.

**Reviewer #1 (Public Review):**
Summary:The manuscript by Ozcan et al., presents compelling evidence demonstrating the latent potential of glial precursors of the adult cerebral cortex for neuronal reprogramming. The findings substantially advance our understanding of the potential of endogenous cells in the adult brain to be reprogrammed. Moreover, they describe a molecular cocktail that directs reprogramming toward corticospinal neurons (CSN).Strengths:Experimentally, the work is compelling and beautifully designed, with no major caveats. The main conclusions are fully supported by the experiments. The work provides a characterization of endogenous progenitors, genetic strategies to isolate them, and proof of concept of exploiting these progenitors' potential to produce a specific desired neuronal type with "a la carte" combination of transcription factors.Weaknesses:Some issues need to be addressed or clarified before publication. The manuscript requires editing. It is dense and rich in details while in other parts there are a few mistakes.

We thank the reviewer for their excellent summary and for their extremely positive review of our paper. We are pleased that the experimental design and conclusions were judged to be wellsupported.

We have revised the paper to enhance clarity, include additional relevant citations, and refine terminology in some sections of the original version.

We appreciate the reviewer’s thoughtful review and agree that these revisions enhance the paper.

**Reviewer #2 (Public Review):**
Summary:Here the authors show a novel direct neuronal reprogramming model using a very pure culture system of oligodendrocyte progenitor cells and demonstrate hallmarks of corticospinal neurons to be induced when using Neurogenin2, a dominant-negative form of Olig2 in combination with the CSN master regulator Fezf2.Strengths:This is a major achievement as the specification of reprogrammed neurons towards adequate neuronal subtypes is crucial for repair and still largely missing. The work is carefully done and the comparison of the neurons induced only by Neurogenin 2 versus the NVOF cocktail is very interesting and convincingly demonstrates a further subtype specification by the cocktail.Weaknesses:As carefully as it is done *in vitro*, the identity of projection neurons can best be assessed *in vivo*. If this is not possible, it could be interesting to co-culture different brain regions and see if these neurons reprogrammed with the cocktail, indeed preferentially send out axons to innervate a co-cultured spinal cord versus other brain region tissue.

We appreciate the reviewer’s positive evaluation of our work and their recognition of its significance in advancing neuronal subtype specification through directed differentiation of endogenous progenitors.

We agree with the reviewer’s suggestion that a very interesting future stage of this work would be to investigate the projection neuron identity *in vivo*. We aim to pursue follow-up studies to investigate *in vivo* integration and connectivity of such neurons generated by directed differentiation from endogenous SOX6+/NG2+ cortical progenitors. As the reviewer insightfully suggests, co-culturing different brain regions with these neurons could offer an alternative strategy to partially assess potential preferential connectivity into cultured spinal cord vs. alternate tissue.

We agree with the reviewer that future investigation *in vivo* will further strengthen the implications of this work.

**Reviewer #3 (Public Review)**:Summary:Ozkan, Padmanabhan, and colleagues aim to develop a lineage reprogramming strategy towards generating subcerebral projection neurons from endogenous glia with the specificity needed for disease modelling and brain repair. They set out by targeting specifically SOX6-positive NG2 glia. This choice is motivated by the authors' observation that the early postnatal forebrain of *Sox6* knockout mice displays marked ectopic expression of the proneural transcription factor (TF) Neurog2, suggesting a latent neurogenic program may be derepressed in NG2 cells, which normally express *Sox6*. Cultured NG2 glia transfected with a construct ("NVOF") encoding Neurog2, the corticofugal neuron-specifying TF Fezf2, and a constitutive repressor form of Olig2 are efficiently reprogrammed to neurons. These acquire complex morphologies resembling those of mature endogenous neurons and are characterized by fewer abnormalities when compared to neurons induced by Neurog2 alone. NVOF-induced neurons, as a population, also express a narrower range of cortical neuron subtype-specific markers, suggesting narrowed subtype specification, a potential step forward for Neurog2-driven neuronal reprogramming. Comparison of NVOF- and Neurog2-induced neurons to endogenous subcerebral projection neurons (SCPN) also indicates Fezf2 may aid Neurog2 in directing the generation of SCPN-like neurons at the expense of other cortical neuronal subtypes.Strengths:The report describes a novel, highly homogeneous *in vitro* system amenable to efficient reprogramming. The authors provide evidence that Fezf2 shapes the outcome of Neurog2-driven reprogramming towards a subcerebral projection neuron identity, consistent with its known developmental roles. Also, the use of the modified RNA for transient expression of Neurog2 is very elegant.Weaknesses:The molecular characterization of NVOF-induced neurons is carried out at the bulk level, therefore not allowing to fully assess heterogeneity among NVOF-induced neurons. The suggestion of a latent neurogenic potential in postnatal cortical glia is only partially supported by the data from the *Sox6* knockout. Finally, some of the many exciting implications of the study remain untested.Discussion:The study has many exciting implications that could be further tested. For example, an ultimate proof of the subcerebral projection neuron identity would be to graft NVOF cells into neonatal mice and study their projections. Another important implication is that *Sox6*-deficient NG2 glia may not only express Neurog2 but activate a more complete neurogenic programme, a possibility that remains untested here.Also, is the subcerebral projection neuron dependent on the starting cell population? Could other NG2 glia, not expressing *Sox6*, also be co-axed by the NVOF cocktail into subcerebral projection neurons? And if not, do they express other (Sox) transcription factors that render them more amenable to reprogramming into other cortical neuron subtypes? The authors state that SOX6-positive NG2 glia are a quiescent progenitor population. Given that NG2 glia is believed to undergo proliferation as a whole, are SOX6-positive NG2 glia an exception from this rule? Finally, the authors seem to imply that subcerebral projection neurons and SOX6-positive NG2 glia are lineage-related. However, direct evidence for this conjecture seems missing.

We appreciate the reviewer’s thoughtful and detailed review of this work. We especially appreciate the positive evaluation of the work and the highlighting of multiple strengths of our approach, including the role of Fezf2 in refining neuronal subtype identity and the use of modified RNA to enable transient expression of Neurog2.

We acknowledge the reviewer’s comment that single-cell transcriptomic analysis would indeed provide a more granular view of likely heterogeneity. This current study focuses on investigating the feasibility of directed differentiation of corticospinal-like neurons from endogenous progenitors. Future work employing single-cell sequencing could indeed help delineate the heterogeneity of neurons generated by directed differentiation, and potentially contribute toward identification of potential molecular roadblocks in different subsets.

Regarding the suggestion that SOX6-deficient NG2+ progenitors might activate a broader neurogenic program, we agree that this is an intriguing possibility. We are currently conducting indepth investigation of the loss of SOX6 function in NG2+ progenitors, and we aim to submit this quite distinct work for separate publication.

The reviewer raises an important point about whether SOX6+/NG2+ progenitors and subcerebral projection neurons are indeed normally lineage-related. In the current work, we utilized postnatal cortical SOX6+/NG2+ progenitors that are thought to be largely derived from EMX1+ and GSH2+ ventricular zone neural progenitors. Our unpublished data from the separate study noted above indicate that SOX6 is expressed by both these lineages *in vivo*. Since subcerebral projection neurons are derived from EMX1+ ventricular zone progenitors (SOX6-expressing), at least some of the SOX6+/NG2+ progenitors are expected to share a lineage relationship with subcerebral projection neurons. While our data strongly suggest such a link, we agree that direct lineagetracing could be pursued in future work.

Finally, we agree with the reviewer’s suggestion that *in vivo* transplantation to assess the identity and connectivity of neurons generated by directed differentiation would be very interesting, and is a natural next phase of this work. We aim to pursue such work in future investigations.

We again thank the reviewer for their insightful comments.

**Reviewer #1 (Recommendations For The Authors):**
The most important clarification for me concerns the initial description of the progenitors. I think there is a mistake with the transgenic line NG2. The dsRed mouse used in Figure 1 C is not described until later in the results describing Figure 2. This was confusing. Moreover, perhaps this is a reason why I get confused and do not understand how the authors conclude that SOX6+ cells are a subset of NG2positive cells. Panel C shows the opposite. Please correct the description and show the quantification of data in panel 1C.

We thank the reviewer for their thoughtful review and for highlighting this important point. We appreciate the reviewer pointing out the benefit of further clarity regarding the NG2.DsRed transgenic mouse description in Figure 1C. We have revised the text to clarify the use of the transgenic line and ensure that the DsRed mouse is properly introduced. Additionally, we have further clarified the description explaining the basis for concluding that SOX6+ cells are a subset of NG2+ cells and further integrate this conclusion with the data presented.

During cell sorting from the cortices of NG2.DsRed mice, we observe two distinct populations of NG2-DsRed+ cells based on fluorescence intensity in FACS: NG2-DsRed “bright” and NG2-DsRed “dim” populations. The NG2-DsRed “dim” population consists of a heterogenous mix of NESTIN+ progenitors, GFAP+ astrocytes/progenitors, a subset of NG2+ cells, and other unidentified cells. In contrast, the DsRed “bright” population includes a broader group of progenitors that also give rise to oligodendrocytes (please see Zhu, Bergles, and Nishiyama 2008), along with pericytes.

Previous studies have shown that, while dorsal/pallial VZ progenitors express SOX6 during embryonic development, SOX6 expression becomes restricted to interneurons postnatally (these do not express NG2 proteoglycan; Azim et al., 2009) and to the broader group of NG2+ progenitors that also give rise to oligodendrocytes. The ICC image in Fig. 1C shows bright NG2+ cells in the cortex, many of which express SOX6. Thus, we conclude that SOX6+ cells constitute a subset of NG2-DsRed+ cells.

In a similar line, the work is beautiful, but the manuscript can gain a lot from shortening and some more editing. for example:(1) In the abstract, the word inappropriate should be removed. It seems to me that is an unnecessary subjective qualification - it is hardly possible that in biology we found repression of something inappropriate.

We have removed the word “inappropriate”.

(2) FACS-purify these genetically accessible....establish a pure culture. Genetically accessible is nice, and I understand that it conveys that they can be traced in the mouse, but everything is genetically accessible with the right tool, and perhaps it is more informative to explain which gene or report is used for the isolation. These cells are not accessible in humans. Also, I consider it best to remove pure- the culture is pure (purified by FACS) cells.

We have revised the text to specify the gene/reporter used for isolation instead of using "genetically accessible", and we removed "pure", since FACS purification is already explicitly mentioned.

(3) In the initial paragraph in the results: "They are exposed to the same morphogen gradients throughout embryonic development, and thus, compared to distant cell types, have similar epigenomic and transcription landscapes." This is proven in the cited publication, but the way is stated here seems a bit of an unnecessary overstatement. The hypothesis stated after this paragraph is as good as it is with or without this argument.

We have revised the text and simplified the statement. We agree that the hypothesis remains clear and well-supported without this emphasis.

(4) In the result sections, "two distinct populations of DsREd-positive cells were identified based on fluorescence intensity"- I know it is correct, but when reading the percentages, I was confused because those percentages divided the population into three fractions. What the authors do not explain is that they discard the intermediate-expressing population.

We appreciate the reviewer highlighting this inadvertent point of confusion. We erred by discussing only the two populations of central interest to us (DsRed-bright and DsRed-dim), and did not explicitly mention the DsRed-negative population. We have now clarified the text to include all three cell populations and their percentages of the total cells in all three populations (in the original manuscript and still now, ~75-78% were DsRed-negative). We have also further clarified that only DsRed-Bright cells (identified as progenitors) were used for all subsequent experiments.

These examples illustrate the type of editing that would be appreciated but which is entirely up to the authors.

We thank the reviewer for their thoughtful suggestions toward improving clarity and precision. We have incorporated these recommendations, along with suggestions from the other two reviewers, in the revised paper.

**Reviewer #2 (Recommendations For The Authors):**
(1) The authors start their results section by showing *in situ* Hybridization for Ngn2 in control and Sox6KO mice. These control sections do not look convincing, as there is not even some signal in the adult VZSVZ region and virtually no background. Please show sections where some positive signal can also be detected in the control sections.

We agree with the reviewer that making direct comparisons in ISH experiments is an important point. In our ISH experiments, to ensure consistency and appropriate comparisons, we process WT and KO sections together and stop the signal development simultaneously. We could have extended the development time to enhance WT signal to a detectable level, but that would have led to excessive background and over-saturated signal in the KO sections.

To address the reviewer’s point, we have added a new supplementary figure with an additional pair of WT and KO sections, along with reference data from the Allen Brain Atlas. The WT section shows faint Neurog2 expression in the dentate gyrus region of the hippocampus, while the KO section confirms very substantial upregulation of Neurog2 in the absence of SOX6 function. These additional data enhance the clarity and depth of our results.

Please see the following link for the Allen Brain Atlas ISH data demonstrating that Neurog2 expression in the postnatal (P4) SVZ/SGZ is inherently low. (https://developingmouse.brainmap.org/experiment/show/100093831).

(2) As a hallmark of projection neurons is where they send their axons, it would be important to include a biological assay for this. Of course, *in vivo* experiments would be great, but if this is not possible, the authors could co-culture sections from the late embryonic cortex, striatum, and spinal cord to see if the reprogrammed neurons preferentially extend their axons towards one of these targets (as normally developing neurons would, see e.g. Bolz et al., 1990).

We agree with the reviewer’s suggestion that a very interesting future stage of this work would be to investigate the projection neuron identity including connectivity *in vivo*. We aim to pursue follow-up studies to investigate *in vivo* integration and connectivity of such neurons generated by directed differentiation from endogenous SOX6+/NG2+ cortical progenitors. As the reviewer insightfully suggests, co-culturing different brain regions with these neurons could offer an alternative strategy to partially assess potential preferential connectivity into cultured spinal cord vs. alternate tissue. This area of investigation is of substantial interest to our lab, and we aim to pursue it in the coming years– it is a very large undertaking by either approach.

(3) However, if the loss of *Sox6* is sufficient for Ngn2 to be upregulated, why did the authors not pursue this approach in their reprogramming experiments? Are these endogenous levels sufficient for reprogramming? Please add some OPC cultures from WT and KO mice to explore their conversion to neurons and possibly combine them with Olig2VP16 and Fezf2.

We thank the reviewer for this insightful comment and for raising this broader area of inquiry regarding whether SOX6 might be down-regulated to enhance induction of neurogenesis. We are writing a separate manuscript regarding function of SOX6 in these progenitors during normal or molecularly manipulated development. We investigate function of SOX6 using both whole body null mice and a series of conditional null mice. We aim to post that work as a preprint and submit it for review and publication in the coming months. Beyond that work, the potential strategy of downregulating SOX6 function while simultaneously upregulating other molecular controls to refine directed neuronal differentiation is also of substantial interest to us, and we aim to pursue this in follow-up work. Though these are both interesting questions/topics, we respectfully submit that these broad areas of parallel, complex, and future investigation would substantially expand the scope of work in this paper, so we aim to address them in separate studies.

(4) Please indicate independent biological replicates as individual data points in all histograms, i.e. also in Figure 2K, Figure 4I, S2H.

We have updated the figure legends indicating the biological replicates, and explained the broad media optimization that was used successfully in all further experiments.

(5) GFP labelling in Figures S2K-N is not convincing - too high background. Please optimize.

We have redesigned this figure and now present it as a new supplementary figure, with GFP pseudocolored in gray and enlarged subpanels for improved visualization of cell morphology.

**Reviewer #3 (Recommendations For The Authors):**
This is an extremely well-written manuscript with very exciting implications. Obviously, not all can be tested here. Some of the suggestions are relatively easy and may be worth testing right away, others may require more extensive study in the future. In my view, completing some of the points below could make this paper a landmark study.I start with the key questions:(1) Do grafted NVOF cells give rise to subcerebral projection neurons *in vivo*?

We agree with the reviewer’s suggestion that a very interesting future stage of this work would be to investigate the projection neuron identity including connectivity *in vivo*. As noted above in response to Reviewer 2, we aim to pursue follow-up studies to investigate *in vivo* integration and connectivity of such neurons generated by directed differentiation from endogenous SOX6+/NG2+ cortical progenitors. This question is of substantial interest to us, and we aim to pursue it in the coming years– as the reviewer notes, this is a very large undertaking, and beyond the scope of this paper.

(2) What is the fate of the *Sox6* deficient NG2 glia that express Neurog2? One could isolate these cells and subject them to scRNA sequencing to see how far neurogenesis proceeds without addition of exogenous factors.

We thank the reviewer for this insightful question. As noted in our response to Reviewer 2, we are writing a separate manuscript regarding function of SOX6 in these progenitors during normal or molecularly manipulated development. We investigate function of SOX6 using both whole body null mice and a series of conditional null mice. We aim to post that work as a preprint and submit it for review and publication in the coming months, likely in early summer. We respectfully submit that this broad area of parallel, complex investigation would substantially expand the scope of work in this paper and make this paper too complex and multi-directional, so we aim to publish them as separate papers for the benefit of clarity for readers.

(3) Obviously, what happens to Sox6-deficient (or non-deficient cells) when forced to express NVOF? In this context, it might be fair to cite Felske et al (PLoS Biol, 2023) who report Neurog2 and Fezf2-induced reprogramming in the postnatal brain. In their model, these authors did not distinguish between converted astrocytes and NG2 glia. Thus, some of the reprogrammed cells may comprise the SOX6positive cells described here.

We thank the reviewer for highlighting for us that we inadvertently omitted referencing the important paper by Felske et al., 2023. We have now included this citation.

We thank the reviewer for raising this broader area of inquiry regarding whether SOX6 might be down-regulated to enhance induction of neurogenesis. Beyond the work noted above regarding function of SOX6 in these progenitors during normal or molecularly manipulated development, the potential strategy of downregulating SOX6 function while simultaneously upregulating other molecular controls to refine directed neuronal differentiation is of substantial interest to us, and we aim to pursue this in follow-up work. We again respectfully submit that this area of complex, future investigation should be addressed in future studies.

Very interesting unaddressed questions include:(1) Are Sox6+ NG glia of dorsal origin? This is implied but not shown. One could use Emx1Cre lines to assess this. Are Sox6+ glia and subcerebral projection neurons clonally related? This may be more challenging. In this context, it might be again fair to refer to Herrero-Navarro et al (Science Advances 2021) who show that glia lineage related to nearby neurons gives rise to induced neurons with regional specificity.

The reviewer raises an important question regarding the competence of SOX6+/NG2+ progenitors from distinct origins to generate corticospinal-like neurons by directed differentiation. In ongoing unpublished work, we have identified SOX6 expression by NG2+ progenitors of the three lineages derived from ventricular zone progenitors that express either Emx1, Gsh2, or Nkx2.1 transcription factors. The EMX1+ lineage-derived SOX6+/NG2+ progenitors are directly lineage related to cortical projection neurons. As the reviewer suggests, future experiments could explore potential differences in competence between these three populations.

We again thank the reviewer for highlighting for us that we also inadvertently omitted referencing the exciting study by Herrero-Navarro that addresses the question of regional heterogeneity within astrocytes and the differential reprogramming potential related to their origins. We have now cited this paper in the manuscript.

(2) Do other NG2 glia not give rise to subcerebral projection neurons when challenged with NVOF? Thus, how important is *Sox6* expression really?

The question of the specific competence of dorsal/cortical SOX6+/NG2+ progenitors to differentiate into corticospinal-like neurons, and the strategy of downregulating SOX6 function while simultaneously upregulating other molecular controls to direct neuronal differentiation, are both of great interest to us. In pilot experiments, we observed reduced competence of ventrallyderived SOX6+/NG2+ progenitors to generate similar neurons. We plan to pursue the SOX6 manipulation in follow up work.

(3) Do Sox6+ NG2 glia proliferate like other NG2 glia and thereby represent a replenishable pool of progenitors?

Yes; as noted in the text shortly after Figure 1, and as presented in Figure S3l-L, these progenitors proliferate robustly in response to the mitogens PDGF-A and FGF2.

(4) How heterogenous are the NVOF-induced neurons? The bulk highlights the overall specificity, but does not tell whether all cells make it equally well.

We agree with the reviewer that this is an interesting question. ICC analysis (Fig. 4G-4H) presents the variation in the levels of a few functionally important proteins in the population of NVOFinduced neurons. This could be due to any or all of at least three potential possibilities: (1) potential diversity in the population of purified SOX6+/NG2+ progenitors; (2) technical variability in the amount of NVOF plasmid delivered to individual progenitors during transfection; and/or (3) natural stochastic TF-level variations generating closely-related neuron types, that also occurs during normal development. Future experiments could explore these questions.